# Topological stress triggers persistent DNA lesions in ribosomal DNA with ensuing formation of PML-nucleolar compartment

**Alexandra Urbancokova[1†], Terezie Hornofova[1†], Josef Novak[1], Sarka Andrs Salajkova[1‡], Sona Stemberkova Hubackova[1§], Alena Uvizl[1#], Tereza Buchtova[2], Martin Mistrik[2], Brian McStay[3], Zdenek Hodny[1]\*, Jiri Bartek[1,4,5]\*, Pavla Vasicova[1]\***

[1]Laboratory of Genome Integrity, Institute of Molecular Genetics of the Czech Academy of Sciences, Prague, Czech Republic; [2]Institute of Molecular and Translational Medicine, Faculty of Medicine and Dentistry, Palacky University Olomouc, Olomouc, Czech Republic; [3]Centre for Chromosome Biology, College of Science and Engineering, University of Galway, Galway, Ireland; [4]Genome Integrity Unit, Danish Cancer Society Research Center, Copenhagen, Denmark; [5]Division of Genome Biology, Department of Medical Biochemistry and Biophysics, Karolinska Institutet, Stockholm, Sweden

**\*For correspondence:**
hodny@img.cas.cz (ZH);
jb@cancer.dk (JB);
pavla.vasicova@img.cas.cz (PV)

[†]These authors contributed equally to this work

**Present address:** [‡]Department of Molecular Mechanisms of Disease, University of Zurich, Zurich, Switzerland; [§]Centre for Experimental Medicine, Institute for Clinical and Experimental Medicine, Prague, Czechia; [#]DIANA Biotechnologies a.s., Vestec, Prague, Czechia

**Competing interest:** The authors declare that no competing interests exist.

**Abstract** PML, a multifunctional protein, is crucial for forming PML-nuclear bodies involved in stress responses. Under specific conditions, PML associates with nucleolar caps formed after RNA polymerase I (RNAPI) inhibition, leading to PML-nucleolar associations (PNAs). This study investigates PNAs-inducing stimuli by exposing cells to various genotoxic stresses. We found that the most potent inducers of PNAs introduced topological stress and inhibited RNAPI. Doxorubicin, the most effective compound, induced double-strand breaks (DSBs) in the rDNA locus. PNAs co-localized with damaged rDNA, segregating it from active nucleoli. Cleaving the rDNA locus with I-PpoI confirmed rDNA damage as a genuine stimulus for PNAs. Inhibition of ATM, ATR kinases, and RAD51 reduced I-PpoI-induced PNAs, highlighting the importance of ATM/ATR-dependent nucleolar cap formation and homologous recombination (HR) in their triggering. I-PpoI-induced PNAs co-localized with rDNA DSBs positive for RPA32-pS33 but deficient in RAD51, indicating resected DNA unable to complete HR repair. Our findings suggest that PNAs form in response to persistent rDNA damage within the nucleolar cap, highlighting the interplay between PML/PNAs and rDNA alterations due to topological stress, RNAPI inhibition, and rDNA DSBs destined for HR. Cells with persistent PNAs undergo senescence, suggesting PNAs help avoid rDNA instability, with implications for tumorigenesis and aging.

## eLife assessment

This **valuable** study asks how Promyelocytic leukemia protein (PML) becomes associated with the nucleoli of cells (PML Nucleolar Associations, PNAs) upon various genotoxic stimuli. Using immunostaining analysis with induced DNA double-strand breaks (DSBs) in rDNA repeats, the authors provide **solid** evidence that PNAs are triggered mostly by the inhibition of topoisomerase and RNA polymerase I, which is augmented by homologous recombination but not by the non-homologous end joining double-strand break repair pathway. The findings have potential implications for a better

understanding of how DNA damage in ribosomal DNA is repaired for genome stability. This paper is of interest to researchers in the fields of nuclear structure and DNA repair.

## Introduction

Promyelocytic leukemia protein (PML) was initially studied in the context of acute promyelocytic leukemia (APL), where specific chromosomal translocation t(15;17) occurs, resulting in the fusion of the PML gene with the retinoic acid receptor alpha (RARα; *de Thé et al., 1990*; *Borrow et al., 1990*). The PML-RARα fusion protein is the primary oncogenic driver of APL (*de Thé et al., 1990*). Subsequent investigations revealed that PML possesses diverse functions primarily associated with various stress response pathways. Notably, the complexity of PML's activities is evident in cancer biology, where it exhibits a dual role as both a tumor suppressor and a tumor promoter, depending on the cellular context and the specific signaling pathways involved (reviewed in *Datta et al., 2020*).

*PML* gene encodes seven splicing isoforms with shared N-terminus and variable C-termini, conferring the individual isoforms with different properties and functions (*Bernardi and Pandolfi, 2007*; *Nisole et al., 2013*; *Jensen et al., 2001*). PML is essential for forming PML-nuclear bodies (PML-NBs), which serve as docking sites, facilitating mutual protein interactions and post-translational modifications (*Van Damme et al., 2010*). By this, PML affects diverse functions, including induction of cell cycle arrest, cellular senescence, apoptosis, anti-viral response, chromatin modification, transcriptional regulation, proteasomal degradation, and metabolic regulation (reviewed in *Guan and Kao, 2015*; *Corpet et al., 2020*). Two functions of PML are tightly linked to homology-directed DSB repair (HDR). First, PML-NBs house several HDR proteins, associate with persistent DNA damage foci, and PML depletion leads to decreased survival upon DNA damage requiring HDR-mediated repair (*Vancurova et al., 2019*; *Attwood et al., 2020*; *Boichuk et al., 2011*; *Yeung et al., 2012*; *Münch et al., 2014*). Second, PML is essential for forming APBs (alternative lengthening of telomeres-associated PML bodies), compartments in cells lacking telomerase, where telomeres are maintained via HDR-based mechanisms (*Yeager et al., 1999*; *Zhang et al., 2021*).

The nucleolus is the membrane-less organelle in the nucleus functionally dedicated to pre-rRNA synthesis and ribosome biogenesis (reviewed in *Pederson, 2011*). In a human cell, there are approximately 300 copies of rDNA genes organized in tandem repeats, consisting of the 47S pre-rRNA (13 kb) and an intergenic spacer (30 kb; *Gonzalez and Sylvester, 1995*). The rDNA arrays are flanked by two mostly heterochromatic regions, a proximal junction on the centromeric side and a distal junction (DJ) on the telomeric side (*Floutsakou et al., 2013*). Although these rDNA arrays are dispersed over the *p*-arms of five acrocentric chromosomes 13, 14, 15, 21, and 22 (*Henderson et al., 1972*), they can form the innermost part of the same nucleolus (*Floutsakou et al., 2013*). Notably, as active transcription of rDNA arrays triggers nucleolus formation, these regions are also called nucleolar organizing regions (NORs). It was reported that DJs presumably serve as NORs' anchors at the border of the nucleolus and fundamentally influence the nucleolar organization (*Floutsakou et al., 2013*; *van Sluis et al., 2020*; *van Sluis et al., 2019*; *Mangan and McStay, 2021*). The main threats of undermining rDNA integrity include: (i) collisions between intensive pre-rRNA transcription and replication, inducing rDNA damage (*Lin and Pasero, 2012*); (ii) the intrinsic predisposition of rDNA repeats to recombination events that can occur in cis and cause gain or loss of rDNA units (*Stankiewicz and Lupski, 2002*; *Carvalho and Lupski, 2016*) and (iii) the localization of several rDNA regions within the same nucleolus that under specific circumstances might induce interchromosomal entanglements and massive rearrangement (*Mangan and McStay, 2021*). It is thought that both major DSB repair pathways, NHEJ and HDR, are involved in the repair of rDNA. The choice of the specific pathway depends on several factors and is still largely enigmatic. One of the hypotheses points out the importance of a damage threshold (reviewed in *Korsholm et al., 2020*). According to this notion, a low amount of rDNA DSBs generated, for example by homing endonucleases I-PpoI or AsiSI, can be repaired quickly by NHEJ inside the nucleolus without the concomitant RNAPI inhibition (*Warmerdam et al., 2016*; *Harding and Greenberg, 2016*). In contrast, longer-persisting rDNA DSBs trigger RNAPI inhibition followed by segregation of rDNA into nucleolar caps, where HDR is the main repair pathway involved (*Warmerdam et al., 2016*; *van Sluis and McStay, 2015*).

Under certain stress conditions, PML can accumulate on the border of the nucleolar cap or form spherical structures containing nucleolar material next to the nucleolus, termed collectively

PML-nucleolar associations (PNAs). PNAs formation was observed in response to different types of genotoxic insults (*Janderová-Rossmeislová et al., 2007*; *Bernardi et al., 2004*; *Shav-Tal et al., 2005*; *Condemine et al., 2007*), upon proteasome inhibition (*Mattsson et al., 2001*), and in replicatively senescent human mesenchymal stem cells (hMSC) and human fibroblasts (*Janderová-Rossmeislová et al., 2007*; *Condemine et al., 2007*). Doxorubicin, a topoisomerase inhibitor and one of the PNAs inducers, provokes a dynamic interaction of PML with the nucleolus, where the different stages of PNA maturation linked to RNAPI inhibition can be discriminated – PML 'bowls', PML 'funnels', PML 'balloons' and PML-nucleolus-derived structures (PML-NDS; *Imrichova et al., 2019*). Using live cell imaging, we observed a dynamic interconnection among these structural subtypes (see scheme *Figure 1A*). The structural transition of doxorubicin-induced PNAs starts with the accumulation of diffuse PML around the nucleolar cap, forming a PML bowl. Note that this event does not occur immediately after induction of genotoxic stress and is coupled to the later stress response as the highest number of PNAs was observed between the first and second day after doxorubicin treatment (*Imrichova et al., 2019*). As the RNAPI inhibition continues, PML bowls can protrude into PML funnels or transform into PML balloons, wrapping the whole nucleolus. When the stress is relieved, and RNAPI resumes activity, a PML funnel transforms into PML-NDS, a distinct compartment placed next to the non-segregated (i.e. reactivated) nucleolus. PML-NDS contains nucleolar material, rDNA, and markers of DSBs (*Imrichova et al., 2019*; *Hornofova et al., 2022*). Recently, we found that a protein region encoded by exon 8b of PML gene and the SUMO-interacting motif (SIM) are the only domains responsible for the recognition of the nucleolar cap and that this association is tightly regulated by p14$^{ARF}$-p53 and CK2 (*Hornofova et al., 2022*). Notably, exon 8b is unique for the PML IV isoform that plays the primary role in the clustering of damaged telomeres (*Zhang et al., 2021*). Interestingly, PML determinants regulating the interaction with the nucleolus (nucleolar cap) are dispensable for the interaction of PML with persistent DSBs after ionizing radiation (IR; *Hornofova et al., 2022*), pointing again to the fine regulation of this interaction.

In the present study, we explored the stimuli responsible for triggering the formation of PNAs, the kinase signaling required for PNAs formation, as well as the links with DNA repair modes and cell cycle stages. As potential triggers of PNAs, we examined factors involved in various cellular processes, including the inhibition of DNA topoisomerase I (TOP1), DNA topoisomerase II (TOP2), RNAPI, replication stress, IR, and proteasome inhibition. This analysis enabled us to identify common features among stimuli leading to the formation of PNAs. Since PNAs were found to associate with markers of the DSB response, we further investigated how the cleavage of rDNA by the endonuclease I-PpoI and the modulation of DNA repair pathways influence the generation of PNAs. Our findings suggest that the induction of PNAs is primarily driven by persistent rDNA damage induced by topological stress and rDNA DSBs, which undergo resection and are directed to HDR or backup repair pathways.

## Results
### Simultaneous inhibition of topoisomerases and RNA polymerase I induces PML-nucleolar interaction

To investigate the nature of signals triggering the formation of PNAs, we treated human telomerase-immortalized retinal pigment epithelial cells (RPE-1$^{hTERT}$) with a range of compounds affecting various cellular processes, including RNAPI, TOP1 and TOP2 inhibitors, DNA-damaging agents, and compounds affecting pre-rRNA processing, etc. (see *Supplementary file 1*). To pinpoint the common features among stimuli that initiate PNAs, we quantified the number of nuclei exhibiting different PNA structural types by staining PAF49 (subunit of RNAPI and marker of nucleolus) and PML after a 48 hr exposure period (different PNA structural types are shown in *Figure 1A* and *Figure 1—figure supplement 1*; quantification of nuclei with PNAs is in *Figure 1B*). This time point was chosen based on our previous observation of a peak in this event following treatment with doxorubicin (*Imrichova et al., 2019*). Concurrently, we monitored several other parameters at the same time point. These included the localization of the RNAPI PAF49 subunit (the spatial segregation of PAF49 and its accumulation into the nucleolar cap serves as an indicator of RNAPI inhibition; for a pattern of segregated and non-segregated PAF49, see *Figure 1—figure supplement 1*), the phosphorylation of serine 139 on histone H2AX (γH2AX foci marking DSBs), the stabilization of p53 protein levels (acting as a signal of DNA damage response (DDR) and cellular stress), and alterations in the levels of TOP1 and DNA

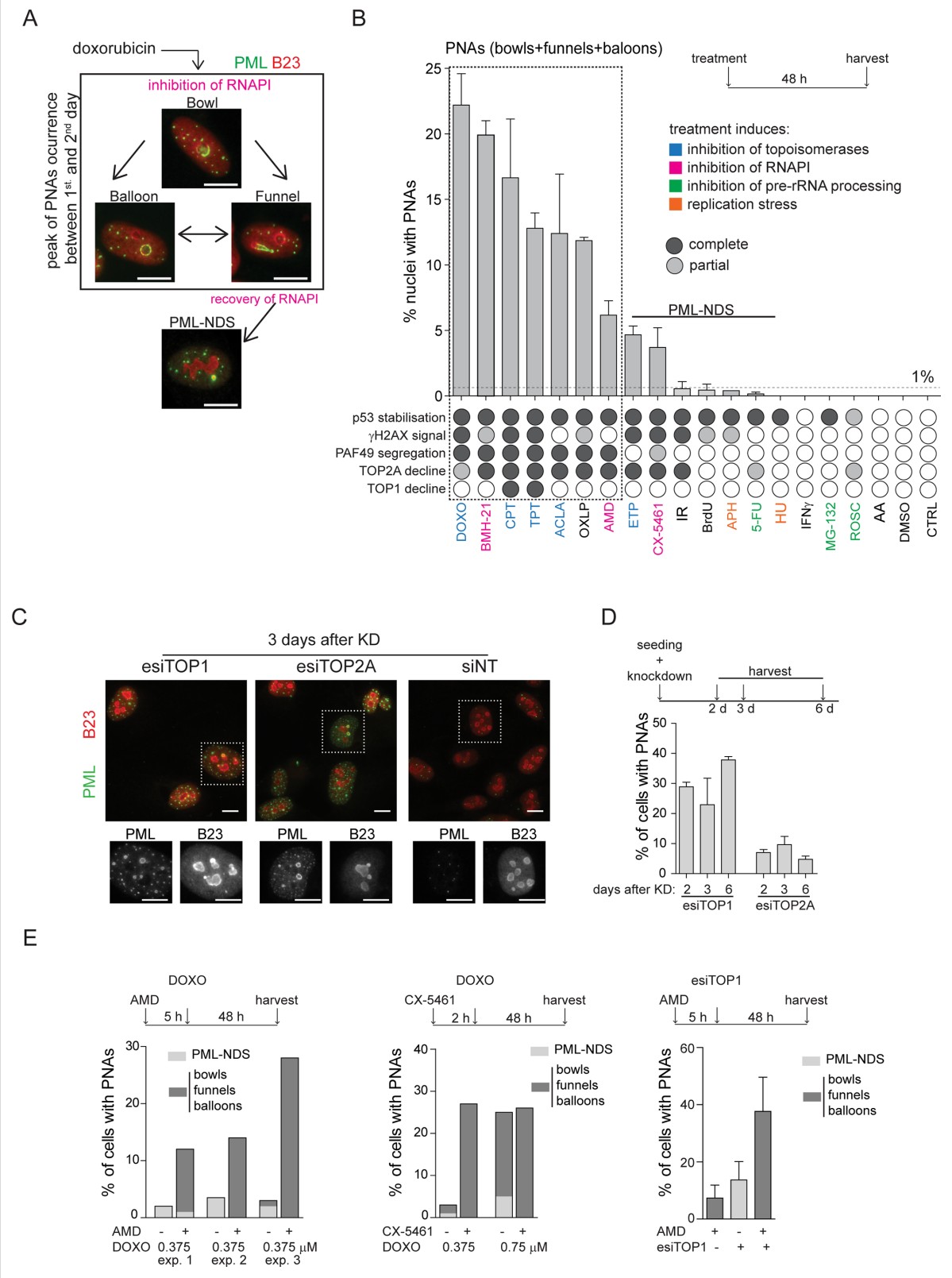

**Figure 1.** PNAs formation after diverse stress-inducing stimuli and topoisomerase downregulation. (**A**) Structural types of PML-nucleolar associations (PNAs) – 'bowls', 'funnels', 'balloons', and PML-nucleolus-derived structures (PML-NDS), occurring in RPE-1$^{hTERT}$ cells after treatment with 0.75 μM doxorubicin detected by indirect immunofluorescence (IF) with anti-PML antibody (green) and anti-B23 (red, for nucleoli visualization). (**B**) Quantifying the percentage of RPE-1$^{hTERT}$ cells containing PNAs, 48 hr after treatment with various stress-inducing stimuli. The stress stimuli were divided into

*Figure 1 continued on next page*

*Figure 1 continued*

five categories according to their mechanism of action (see ***Supplementary file 1***): (1) poisons/inhibitors of topoisomerases, (2) treatments inducing the inhibition of RNAPI, (3) inhibitors of pre-RNA processing, (4) inductors of replication stress, and (5) other stressors. p53 stabilization, γH2AX foci formation, PAF49 segregation, and TOP1 or TOP2A decline have been assessed for each treatment. The concentration and abbreviation used: DOXO, doxorubicin (0.75 μM); BMH-21 (0.5 μM); CPT, camptothecin (50 μM); TPT, topotecan (50 μM); ACLA, aclarubicin (0.05 μM); OXLP, oxaliplatin (10 μM); AMD, actinomycin D (10 nM); ETP, etoposide (50 μM); CX-5461 (5 μM); IR, (ionizing radiation 10 Gy); APH, aphidicolin (0.4 μM); 5-FU, 5-fluorouracil (200 μM); HU, hydroxyurea (100 μM), IFNγ (5 ng/mL); MG-132 (10 μM); ROSC, roscovitine (20 μM); AA, acetic acid. The mean ± SEM from at least two biological replicates is shown. (**C**) The pattern of PML (green) and B23 (red) in RPE-1$^{hTERT}$ cells visualized by indirect IF, 3 days post-transfection with siRNAs targeting TOP1 and TOP2A, or with non-targeting siRNA (siNT), respectively. (**D**) Quantification of the percentage of RPE-1$^{hTERT}$ cells containing PNAs 2, 3, and 6 days after transfection with esiTOP1 and esiTOP2A. The mean ± SEM from at least two biological replicates is shown. (**E**) RPE-1$^{hTERT}$ cells were pre-treated with 10 nM AMD for 5 hours or with 5 μM CX-5461 for 2 hours to inhibit RNAPI. The cells were then treated with 0.375 μM or 0.75 μM doxorubicin or transfected with esiRNA targeting TOP1 for 48 hr. The bar graphs show the percentage of cells containing either PML-NDS or bowls/funnels/balloons for three biological replicates (graph 1), for one biological replicate (graph 2), or the mean ± SEM for three biological replicates (graph 3). Scale bar, 10 μm.

The online version of this article includes the following source data and figure supplement(s) for figure 1:

**Source data 1.** Numerical data for B, D, and E.

**Figure supplement 1.** Detection of RNAPI segregation after various genotoxic treatments.

**Figure supplement 2.** Detection of γH2AX signal after various genotoxic treatments.

**Figure supplement 2—source data 1.** Numerical data for whiskers plots.

**Figure supplement 3.** Detection of p53, TOP2A and TOP1 after various genotoxic treatments.

**Figure supplement 3—source data 1.** RAW membranes for A, B, and C.

**Figure supplement 3—source data 2.** Uncropped and labeled membranes for A, B, and C.

**Figure supplement 4.** PNAs after treatments affecting functions of topoisomerases and RNAPI.

**Figure supplement 4—source data 1.** RAW membranes for D.

**Figure supplement 4—source data 2.** Uncropped and labeled membranes for D.

**Figure supplement 4—source data 3.** Numerical data for A, B, and G.

**Figure supplement 5.** Detection RNAPI inhibition after AMD and CX-5461 treatment.

topoisomerase II alpha (TOP2A; for detection and quantification of these parameters see ***Figure 1— figure supplements 1–3*** and ***Figure 1B***).

The treatment of hydroxyurea and aphidicolin (DNA replication inhibitors), 5-FU (DNA/RNA synthesis and pre-rRNA late processing inhibitor), MG-132 (proteasome and pre-rRNA late processing inhibitor), roscovitine (CDK and pre-rRNA early processing inhibitor), IR (for DNA damage), IFNγ (PML expression inducer), and thymidine analog bromodeoxyuridine (BrdU, senescence inducer) did not cause spatial segregation of PAF49 and formation of bowl-, funnel-, or balloon-like PNAs. Occasionally, the PML-NDS, the specific variant of PNAs present next to the non-segregated nucleolus, was formed (<1% of cells; ***Figure 1B***; for characterization of PML-NDS, see also ***Figure 1A*** and ***Figure 1—figure supplement 1***), indicating that stimuli leading to sole stabilization of p53 (***Figure 1—figure supplement 3A***) or induction of PML expression are insufficient to signal for induction of these structures.

In contrast, inhibitors of RNAPI and DNA topoisomerases were potent inducers of PNAs (***Figure 1B***). Further evaluation of the data demonstrated that the propensity for PNAs formation was highest (>5% of cells) for compounds that simultaneously caused the inhibition of RNAPI and TOP2A decline (***Figure 1B***, '***Figure 1—figure supplements 1 and 3B***). These treatments mainly induced the bowl-, funnel-, and ballon-like PNAs (***Figure 1—figure supplement 4A***), reflecting the RNAPI inhibition. For instance, the TOP2 inhibitors doxorubicin and aclarubicin, and the TOP1 inhibitors camptothecin (CPT) and topotecan (TPT) inhibited RNAPI (as indicated by PAF49 spatial segregation, see ***Figure 1—figure supplement 1***), which induced the degradation of TOP2A, generated PNAs in 22%, 12%, 17%, and 13% of cells, respectively. Notably, upon exposure to TOP1 inhibitors (CPT and TPT), only the highest tested concentration (50 μM) induced segregation of PAF49 and a high number of nuclei with PNAs (see ***Figure 1—figure supplement 4B and C***), indicating that the signal necessary for the emergence of PNAs is concentration dependent. The requirement for concurrent inhibition of RNAPI and TOP2A activity to induce the highest number of PNAs was further highlighted by treatment with oxaliplatin and etoposide. Oxaliplatin, primarily reported as a DNA cross-linker, exhibited a combined effect on RNAPI inhibition and TOP2A downregulation, resulting in a high number of

PNAs-containing nuclei (12%). On the other hand, etoposide, a TOP2 poison, did not inhibit RNAPI even at a concentration of 50 µM and induced PNAs (predominantly in the form of PML-NDS) in less than 5% of cells (see *Figure 1B* and *Figure 1—figure supplement 4A, B and C*). We also tested three different compounds that can inhibit RNAPI: actinomycin D (AMD), BMH-21, and CX-5461. All treatments induced a decline of TOP2A but not TOP1 protein level. Notably, after a 2-day-long treatment, only BMH-21 and AMD robustly inhibited RNAPI, whereas CX-5461 did not. BMH-21 was one of the most potent inducers of PNAs, with 20% of cells exhibiting this structure after treatment. In contrast, AMD and CX-5461 induced a comparable number of cells with PNAs (6% and 4%, respectively), ranking these treatments among weaker inducers of PNAs. The differences among the RNAPI inhibitors can be explained by a recent observation that BMH-21 can trap TOP2A and TOP2B (DNA topoisomerase II beta), causing not only RNAPI inhibition but also cumulative topological defects (*Espinoza et al., 2024*). Furthermore, despite the high number of cells with PNAs in the population, BMH-21 and aclarubicin evoked a low amount of γH2AX foci (see *Figure 1—figure supplement 2*), indicating that the extent of DSBs is not the primary signaling factor for the formation of PNAs. In the same vein, IR, etoposide, and CX-5461 exposure induced high levels of γH2AX foci without concomitant occurrence of higher numbers of cells with PNAs, which, in all cases, remained below the 5% of the cell population. These findings suggest that simultaneous impairment of TOP2A/topoisomerases and RNAPI activity enhances the signal for PNAs formation. Notably, the level of DSBs per se does not closely correlate with the stimulation of the PML-nucleolar interaction.

To investigate whether the signal for PNAs formation is directly linked to the abrogation of topoisomerase function and to exclude the possibility of non-specific effects of topoisomerase inhibitors, we downregulated TOP1, TOP2A, and TOP2B, respectively, by RNA interference. Downregulation (knockdown; KD) of TOP1 or TOP2A, but not TOP2B (see *Figure 1—figure supplement 4D* for KD efficiency) induced the formation of PNAs (*Figure 1C and D*, and *Figure 1—figure supplement 4E*). Note that two different siRNAs targeting TOP2B were used. TOP1 and TOP2A were downregulated using a heterogeneous pool of biologically prepared siRNAs (esiRNA; esiTOP1 and esiTOP2A). Following transfection with esiTOP1, we observed that 28%, 23%, and 38% of nuclei exhibited PNAs at 2-, 3-, and 6-days post-transfection, respectively. The KD of TOP2A also prompted the interaction of PML with the nucleolus, albeit only in 7%, 10%, and 5% of cells at the corresponding time points. Interestingly, we noticed that the deficiency of TOP1 at later stages resulted in the inhibition of RNAP.

I. Specifically, on the second day post-TOP1 downregulation, RNAPI remained active, and only PML-NDS were present. However, at later time points (3 and 6 days), RNAPI gradually became segregated, and different types of PNAs, such as funnels and bowls, emerged (*Figure 1—figure supplement 4F and G*). These results suggest that the signal for PNAs is linked to the loss of topoisomerase function, mainly due to the TOP1 deficiency.

Based on the above findings, we proposed that PNA generation reflects a change in DNA metabolism resulting from the combination of actively ongoing pre-rRNA transcription and concomitantly reduced topoisomerase activity. To test this hypothesis, we inhibited RNAPI activity using AMD or CX-5461 before adding doxorubicin or downregulating TOP1 (esiTOP1). The efficacy of RNAPI inhibition was confirmed by 5-FU incorporation (*Figure 1—figure supplement 5*). Surprisingly, in both cases, the cessation of pre-rRNA transcription before inhibition/downregulation of topoisomerases led to more PNAs than treatment with doxorubicin or TOP1 downregulation alone (*Figure 1E*). These results suggest that ongoing pre-rRNA transcription and subsequent treatment-induced collisions are not necessary prerequisites for inducing PNAs.

Overall, our data indicate that defects in topoisomerase activity associated with the inhibition of RNAPI, but not collisions with ongoing rRNA transcription, represent the most potent signals for the formation of PNAs.

## Induction of PNAs is impacted by the inhibition of specific DNA repair pathways

After analyzing these and our previous data (*Imrichova et al., 2019*; *Hornofova et al., 2022*), we argued that persistent (r)DNA damage is likely responsible for prolonged RNAPI inhibition, leading to the PNAs formation. To verify this assumption, we examined whether modifying DNA repair pathways could impact the development of PNAs caused by doxorubicin. First, we inhibited HDR in RPE-1$^{hTERT}$ cells with B02, a compound that blocks RAD51 filament control of homologous strand displacement

(inhibitor of RAD51; *Huang et al., 2011*). Initially, we tested how treatment with only B02 affects PML localization, and we did not observe the formation of PNAs and DSBs (see *Figure 2—figure supplement 1A and B*). Then, we applied a 0.75 µM concentration of doxorubicin, which generated the highest number of PNAs (*Figure 1B*), in combination with increasing doses of B02 (5, 10, and 20 µM; see also *Figure 2—figure supplement 1C and D* for the effect of B02 on RAD51 foci formation). Although the total number of PNAs was not significantly altered, we observed an increased proportion of balloon-like PNAs (*Figure 2—figure supplement 1E– G*). However, the combined treatment caused increased cell death, preventing further follow-up during recovery. Therefore, we also tested the combination of B02 with lower concentrations of doxorubicin (0.375 µM and 0.56 µM; *Figure 2A–D*). Co-treatment of doxorubicin with 20 µM B02 for 48 hr significantly increased the number of cells with PNAs compared to doxorubicin alone (*Figure 2A*) and shifted the proportions of specific PNAs subtypes (*Figure 2B*) towards the forms linked with RNAPI inhibition and nucleolar cap formation (*Imrichova et al., 2019*). These effects were concentration-dependent. While 0.375 µM doxorubicin alone induced predominantly formation of PML-NDS, adding 20 µM B02 resulted in a higher occurrence of bowl-, balloon-, and funnel-like PNAs. This shift towards the balloon-like PNAs linked with the disappearance of PML-NDS was even more augmented by the higher dose of doxorubicin (0.57 µM) combined with 20 µM B02.

Next, we analyzed the effect of HDR inhibition by B02 on the distribution of PNAs in the recovery phase, that is 4 days after doxorubicin removal when the last stage of PNAs, PML-NDS, prevails (*Figure 2E–G*). Adding 20 µM B02 to 0.56 µM and 0.375 µM doxorubicin significantly increased the number of PML-NDS-containing nuclei (as shown in *Figure 2E*). This result indicates that inhibiting RAD51/HDR during doxorubicin treatment affects the dynamics of PNAs during the initial exposure period (2 days) and the recovery phase, likely due to defects in DNA break repair by HDR.

To investigate the contribution of NHEJ in the formation of doxorubicin-induced PNAs, we utilized an inhibitor of DNA-dependent protein kinase (DNA PK; NU-7441) to block NHEJ (*Leahy et al., 2004*). NU-7441 in combination with three concentrations of doxorubicin (0.375, 0.56, and 0.75 µM) did not significantly affect the proportion of nuclei with PNAs (*Figure 2—figure supplement 1H and I*; for the effect of 1 µM NU-7441 on efficiency of DSB repair, see *Figure 2—figure supplement 2A and B*), indicating that the inhibition of NHEJ is not associated with the formation of doxorubicin-induced PNAs.

The results obtained from the previous experiment are consistent with published observations that doxorubicin-induced DNA damage is repaired preferentially by HDR (*Alagpulinsa et al., 2014*; *Schürmann et al., 2021*), and inhibition of this repair pathway caused elevation of nuclei with PNAs. Next, we employed etoposide to investigate whether similar effects could be observed after DNA damage that is preferentially repaired by different pathways. Repair of etoposide-induced DNA damage depends on tyrosyl-DNA phosphodiesterase 2 (TDP2), an enzyme that removes trapped TOP2 from the DNA end (*Cortes Ledesma et al., 2009*). To inhibit such repair, we downregulated TDP2 in RPE-1$^{hTERT}$ cells after etoposide treatment by RNA interference (*Figure 2—figure supplement 2C*) and analyzed PML distribution by indirect IF. While RNAPI was not inhibited and only PML-NDSs were present in nuclei (*Figure 2—figure supplement 2D and E*), the number of nuclei with PML-NDS significantly increased (*Figure 2H*), suggesting that persistent DNA damage can be the source of PNAs – PML-NDS.

These findings showed that inhibiting RAD51 filament formation during doxorubicin treatment and downregulating TDP2 during etoposide treatment induces more PNAs. Thus, we assume that persistent (r)DNA damage augmented by restriction of DNA repair generates a signal for the development of the PML-nucleolar associations.

## Doxorubicin treatment induces PML wrapping around damaged rDNA loci and distal junctions of acrocentric chromosomes

It has been previously demonstrated that PNAs induced by doxorubicin co-localize with rDNA, rDNA-interacting proteins, and DNA damage markers (*Imrichova et al., 2019*; *Hornofova et al., 2022*). Since inhibiting DNA repair pathways after treatment with doxorubicin and etoposide altered PNAs formation, we hypothesized that the generation of PNAs is associated with direct damage to the rDNA locus rather than with general genomic DNA damage. To test this hypothesis, we conducted an immuno-FISH experiment to examine the co-localization of rDNA with a marker of DSBs. Furthermore, to obtain a better understanding of the localization of rDNA loci in the chromosomal context and to discriminate individual nucleolar caps, we used both, rDNA probes and also probes that hybridize with

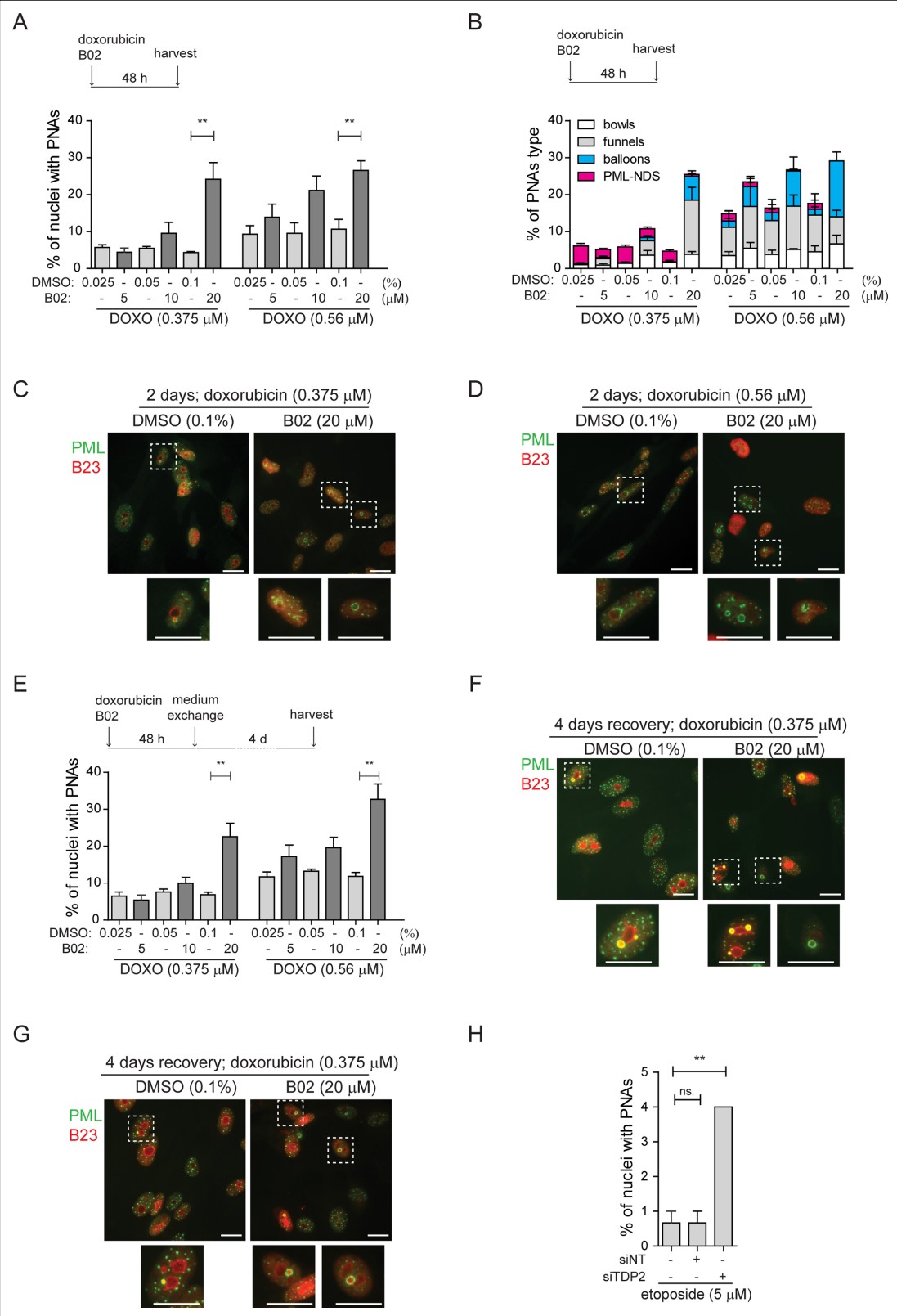

**Figure 2.** Inhibition of DNA repair augmented the PNAs formation. RPE-1[hTERT] cells were treated with doxorubicin and three concentrations of B02 or with etoposide after downregulation of TDP2 by RNA interference. After the treatment, the PML (green) and nucleolar marker B23 (red) were visualized by indirect IF and wide-field fluorescent microscopy, and percentage of nuclei with PNAs was calculated. (**A**) The bar graph represents the percentage of nuclei with PNAs after 2-day-long treatment with doxorubicin (0.375 μM or 0.56 μM), three concentrations of B02 (5, 10, and 20 μM), and corresponding

*Figure 2 continued on next page*

*Figure 2 continued*

concentrations of DMSO as a mock. (**B**) The bar graph represents the distribution of individual types of PNAs after the same treatments as shown in (**A**). (**C**) Representative cells after 2-day-long treatment with 0.375 μM doxorubicin combined with 20 μM B02 or 0.1% DMSO (mock). (**D**) Representative cells after 2-day-long treatment with 0.56 μM doxorubicin combined with 20 μM B02 or 0.1% DMSO (mock). (**E**) The bar graph represents the percentage of nuclei with PNAs after 4 days of recovery from 2-day-long treatments with doxorubicin (0.375 μM or 0.56 μM) together with three concentrations of B02 (5, 10, and 20 μM), and corresponding concentrations of DMSO. (**F**) Representative cells after 4 days of recovery from 2-day-long treatment, with 0.375 μM doxorubicin combined with 20 μM B02 or 0.1% DMSO (mock). (**G**) Representative cells after 4 days of recovery from 2-day-long treatment with 0.56 μM doxorubicin combined with 20 μM B02 or 0.1% DMSO (mock). (**H**) The bar graph represents the percentage of nuclei with PNAs after 2-day-long treatment with 5 μM etoposide in cells where TDP2 was downregulated by RNA interference. In all experiments, at least three biological replicates were evaluated. Results are presented as a mean ± SEM. An unpaired (**A and E**) and paired (**H**) two-tailed t-test were used for statistical evaluation. Asterisks indicate the following: ****p<0.0001, ***p<0.001, **p<0.01, *p<0.05. Scale bar, 20 μm.

The online version of this article includes the following source data and figure supplement(s) for figure 2:

**Source data 1.** Numerical data for A, B, E, and H.

**Figure supplement 1.** The inhibition of specific DNA repair pathways augmented the formation of PNAs.

**Figure supplement 1—source data 1.** Numerical data for A, C, E, F, H, and I.

**Figure supplement 2.** The inhibition of specific DNA repair pathways augmented the formation of PNAs.

**Figure supplement 2—source data 1.** RAW membranes for C.

**Figure supplement 2—source data 2.** Uncropped and labeled membranes for C.

**Figure supplement 2—source data 3.** Numerical data for A.

DJ, a region situated on the telomeric end of the rDNA repeats and previously used as a marker of individual NORs (see the scheme in *Figure 3A*; *Floutsakou et al., 2013*).

First, we examined the localization of rDNA, DJ, PML, and the nucleolar marker B23 in control cells 2 days after doxorubicin treatment and 1 day after doxorubicin removal. As shown in *Figure 3— figure supplement 1* for untreated cells, the rDNA signal was spread throughout the nucleolus, and the DJ signal was detected on the nucleolar rim as expected (*Floutsakou et al., 2013*). After doxorubicin exposure, nucleolar caps emerged, and some of them were wrapped by PNAs (*Figure 3— figure supplement 1*). In several instances, a single PNA (funnel- or bowl-type) appeared to interact with several DJs – acrocentric chromosomes, bridging several nucleolar caps. Twenty-four hours after doxorubicin removal, the PML-NDS emerged, and DJ or rDNA were observed inside or on the rim of these PNAs types (*Figure 3—figure supplement 1*). The co-localization between rDNA/DJ and PML in individual nucleoli was determined using the Fiji/Mosaic/Squassh plugin (*Rizk et al., 2014*). Note that the co-localization analysis was also performed in the nucleoli in which PML was present only in the form of canonical PML-NBs (see the gallery of nucleoli in *Figure 3—figure supplement 2A*). As manifested in *Figure 3—figure supplement 2B*, we proved that both rDNA and DJ co-localized with PML in response to doxorubicin treatment. However, the co-localization coefficient was higher in nucleoli containing PNAs, indicating that the interaction between rDNA/DJ and PML is mainly realized in the form of PNAs.

To demonstrate that doxorubicin induces damage in the form of DSBs in rDNA/DJ, we performed immuno-FISH to detect the 53BP1 foci, a marker of DSBs, together with PML, rDNA, and DJ in control cells and at three different time points after doxorubicin treatment (i.e. 2 days of doxorubicin exposure and 1- and 4 days after doxorubicin removal, respectively; *Figure 3B*). First, we performed a co-localization analysis for rDNA/DJ and PML to verify our previous findings on a different sample set. As shown in *Figure 3C*, we confirmed that doxorubicin treatment induced the co-localization between rDNA/DJ and PNAs. Notably, this co-localization was still detectable even 4 days after doxorubicin removal. We then used the same set of nucleoli to determine the extent of co-localizations between 53BP1 and rDNA/DJ. As shown in *Figure 3D*, we detected the presence of DSBs in both the rDNA locus and DJ region. These findings indicate that the extent of co-localization decreased during the recovery phase. Furthermore, the analysis revealed that rDNA/DJ loci co-localized with 53BP1 even in nucleoli without the PNAs, although the extent of co-localization in such nucleoli was lower. To investigate whether PNAs interact with the rDNA/DJ regions stained for DSBs (53BP1 foci), we utilized the Squassh analysis to combine 53PB1-rDNA/DJ and PML-rDNA/DJ co-localization (for a detailed description of the analysis, see Materials and methods). *Figure 3E* and *Figure 3—figure supplement 3A and B* show an example of the segmentation, quantified overlay, and a 3D model of

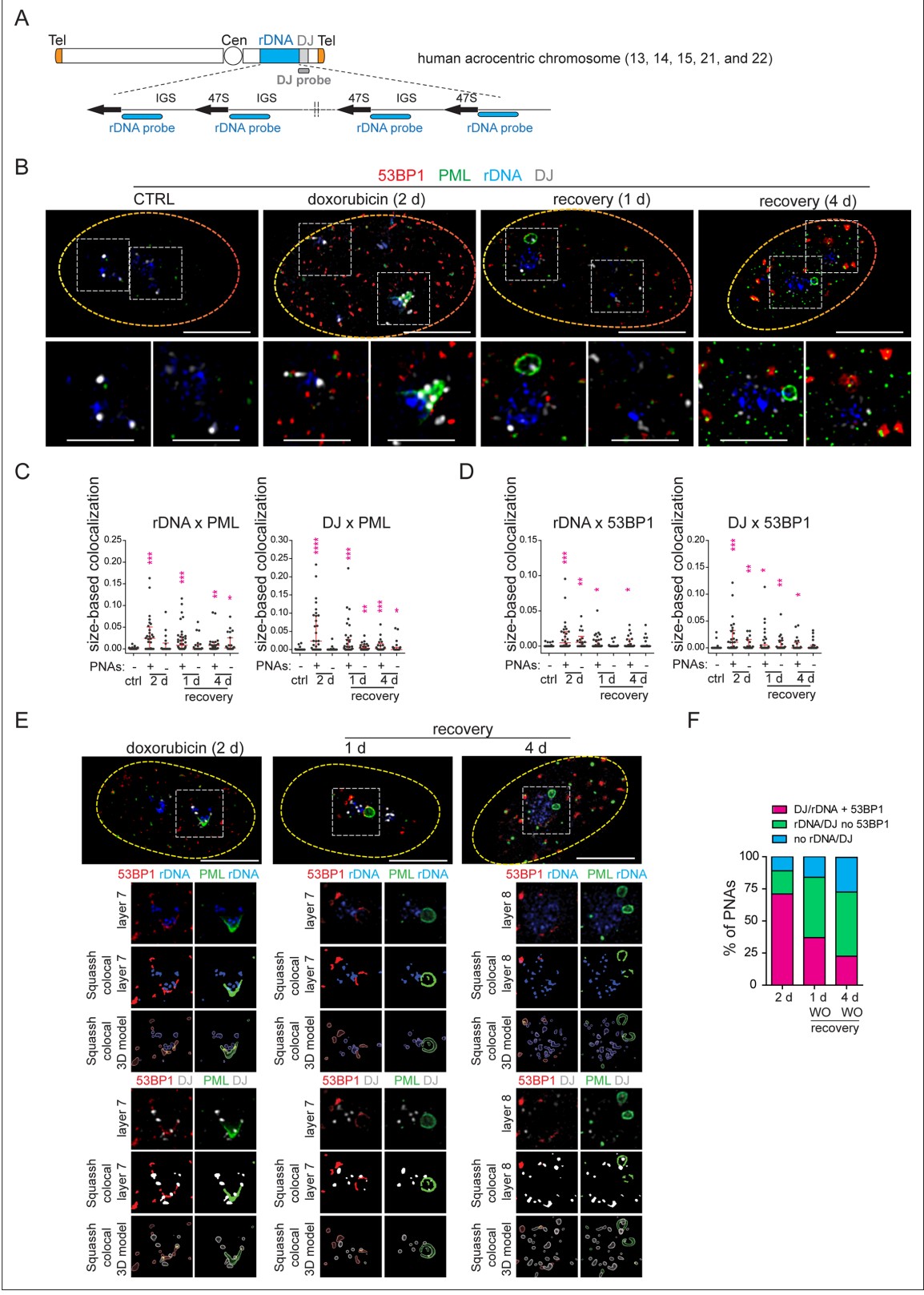

**Figure 3.** PNAs encircle rDNA and DJ loci containing DSB after doxorubicin treatment. RPE-1$^{hTERT}$ cells were treated with 0.75 µM doxorubicin for 2 days and recovered for 1 and 4 days. The proliferating cells were used as a control. rDNA, DJ, PML, and 53BP1 localization was analyzed using immuno-FISH staining and confocal microscopy. (**A**) The scheme of a human acrocentric chromosome. The probes' position for detecting the rDNA locus (blue) and DJ locus (grey) is shown. (**B**) The representative nuclei and the nucleoli with and without PNAs in control and treated cells are shown. rDNA (blue), DJ

*Figure 3 continued on next page*

eLife Research article

Cancer Biology | Cell Biology

*Figure 3 continued*

(white), PML (green), and 53BP1 (red). (**C**) The extent of PML-rDNA and PML-DJ size-based co-localization calculated for individual nucleoli of treated and untreated cells with respect to the presence of PNAs is shown as a scatter plot. The median with an interquartile range is shown. The co-localization was calculated using Fiji(ImageJ)/Mosaic/Segmentation/Squassh plugin. The number of analyzed nucleoli in each group was: ctrl (n=26); 2 days+PNAs (n=28); 2 days without PNAs (n=18); 1-day-long recovery +PNAs (n=38); 1-day-long recovery without PNAs (n=20); 4-day-long recovery +PNAs (n=23); 4-day-long recovery without PNAs (n=24). (**D**) The extent of 53BP1-rDNA and 53BP1-DJ size-based co-localization calculated for individual nucleoli of treated and untreated cells with respect to the presence of PNAs is shown as a scatter plot. The median with interquartile range is shown. The same collection of nucleoli was used as presented in (**C**). (**E**) The example of analysis that was used to identify whether rDNA/DJ with DSB co-localized with PNAs. The images of the representative nucleoli (2 days doxorubicin, 1- and 4- days recovery) after deconvolution, segmentation, and its 3D model are shown. Note, that the signal of all observed markers was identified as a unique 3D object. rDNA (blue), DJ (white), PML (green), and 53BP1 (red). (**F**) The combined bar graph shows the percentage of PNAs containing rDNA/DJ with DSB, rDNA/DJ without DSB, and PNAs in which the rDNA/DJ signal was not detected. The number of analyzed nucleoli: 2 days doxorubicin (n=28), 1-day-long recovery (n=39), 4-day-long recovery (n=21). Non-parametric two-tailed Mann-Whitney test was used for statistical evaluation (**C and D**). Asterisks indicate the following: ****p<0.0001, ***p<0.001, **p<0.01, *p<0.05. Scale bars, 10 µm (nuclei in **B** and **E**) and 5 µm (nucleoli in **B**).

The online version of this article includes the following source data and figure supplement(s) for figure 3:

**Source data 1.** Numerical data for C, D, and F.

**Figure supplement 1.** PNAs envelop rDNA and DJ loci containing rDNA DSB after doxorubicin treatment.

**Figure supplement 2.** PNAs envelop rDNA and DJ loci containing rDNA DSB after doxorubicin treatment.

**Figure supplement 2—source data 1.** Numerical data for B.

**Figure supplement 3.** PNAs envelop rDNA and DJ loci containing rDNA DSB after doxorubicin treatment.

**Figure supplement 3—source data 1.** Numerical data for B.

co-localization. Furthermore, an x/y-scatter plot is given to demonstrate the size of co-localization of rDNA (DJ) objects with 53BP1 foci (axis x) and PML (axis y). Using this approach, we found that about 71% of PNAs examined 2 days after doxorubicin treatment overlapped with rDNA or DJ-containing DSBs marked by 53BP1 foci (*Figure 3F*).

To summarize, our data demonstrate that doxorubicin induces DSBs in the rDNA and DJ regions, which are associated with PML, thereby linking the rDNA damage in the short arm of acrocentric chromosomes with the induction of PNAs. Furthermore, our findings suggest that rDNA/DJ regions with DSBs are enveloped into PML-NDS following the reactivation of RNAPI. Through this mechanism, the damaged rDNA loci become spatially separated from the active nucleolus.

## Direct rDNA damage induced by endonuclease I-PpoI triggers the formation of PML-NDS

Our findings so far indicate that PNAs are linked to rDNA damage. To validate this hypothesis, we used DNA homing-endonuclease I-PpoI to generate targeted breaks in rDNA, as the I-PpoI-induced cleavage site is located inside the rDNA locus in the 28S region (see the scheme in *Figure 4A*; *Stoddard, 2005*; *Berkovich et al., 2007*). To control the activity of I-PpoI, we generated human RPE-1$^{hTERT}$ single-cell clones with a regulatable expression of I-PpoI using the TRE3 GS promoter and destabilization domain FKBT (*Banaszynski et al., 2006*). In preliminary experiments, we analyzed the presence of PNAs after 3-, 6-, 8-, 12-, and 24 hr I-PpoI induction and found that the PNAs were only induced after the 24-hr-long expression of I-PpoI (*Figure 4—figure supplement 1*). In the next experiments, two clones (1A11 and 1H4) were used, and the I-PpoI was activated for 24 hr, followed by the examination of several parameters at different time points (see experimental scheme in *Figure 4B*). As shown in *Figure 4C* and *Figure 4—figure supplement 2A and B*, activating I-PpoI for 24 hr resulted in the appearance of 53BP1 and γH2AX foci on the border of the nucleolus, indicating the presence of (r)DNA breaks. To characterize the extent of (r)DNA damage and the dynamics of its repair, we quantified the number of 53BP1 foci at different time points using high-content microscopy. Nearly 90% of control cells showed no detectable (r)DNA damage, proving that the bulk of DNA damage detected is linked explicitly to I-PpoI activity (*Figure 4D*). Twenty-four hours after I-PpoI expression, the number of 53BP1 foci peaked and then gradually declined over the 5-day recovery phase (*Figure 4D*).

PML localized predominantly to regular PML-NBs, but PML structures containing nucleolar material marked by TOTO-3, resembling the PML-NDS, emerged in about 10% of the nuclei (*Figure 4C and E*). Strikingly, the PNAs specifically associated with nucleolar caps (i.e. bowl-, funnel-, and balloon-type)

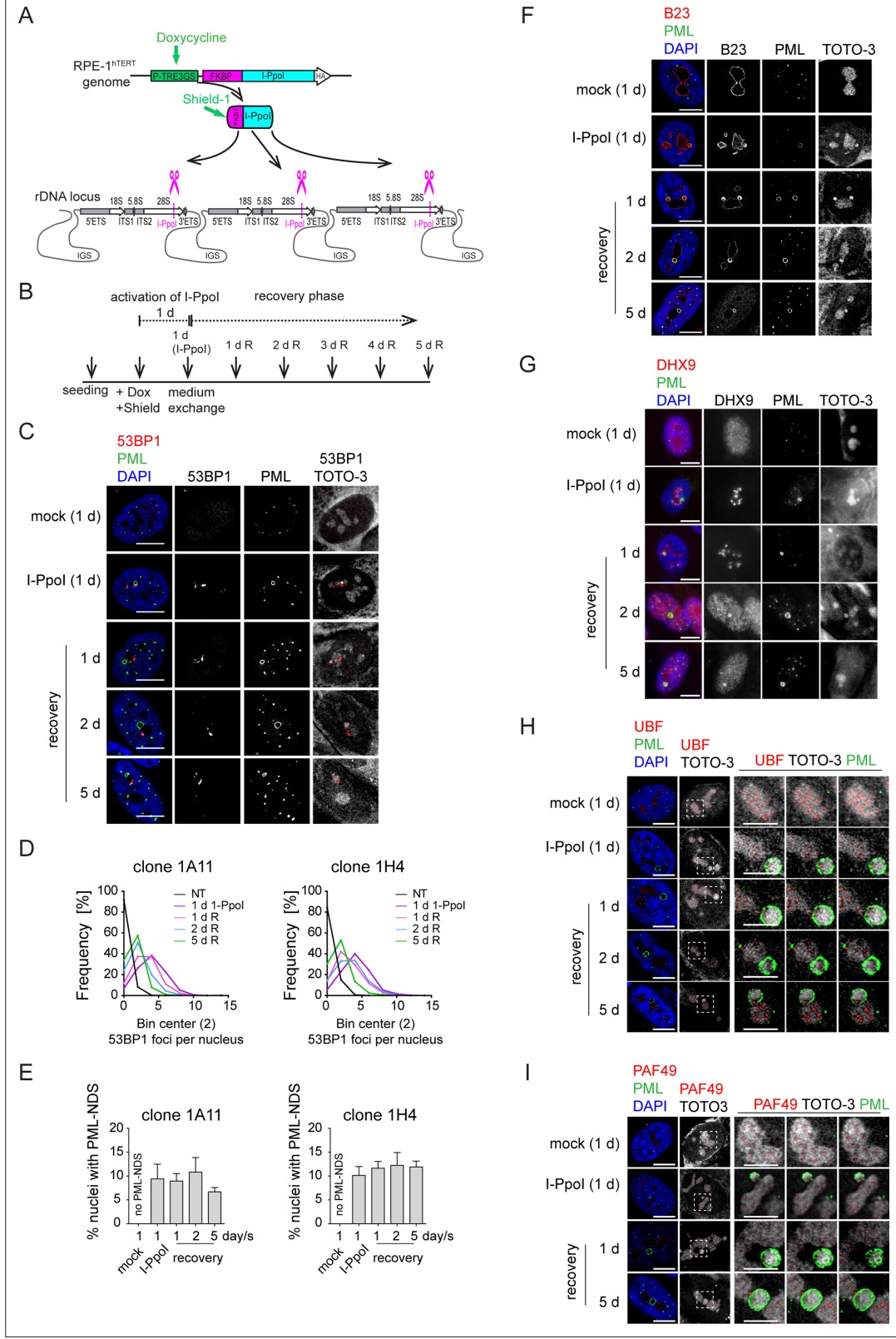

**Figure 4.** The DNA damage introduced into the rDNA locus by endonuclease I-PpoI induces PML-NDS.
(**A**) Scheme of inducible expression of endonuclease I-PpoI in RPE-1$^{hTERT}$ cells and the position of I-PpoI cleavage site in rDNA locus. Tetracycline-inducible promoter (P-TRE3GS), destabilization domain (FKBP), intergenic spacer (IGS), external transcribed spacer (ETS), internal transcribed spacer (ITS). (**B**) Scheme of the experimental

*Figure 4 continued on next page*

*Figure 4 continued*

setup used in all experiments presented. Briefly, I-PpoI was activated by doxycycline and Shield for 24 hr, then the medium was exchanged, and the cells were analyzed upon the recovery phase (0, 1, 2, and 5 days). (**C**) The representative images obtained by indirect IF and confocal microscopy (STELLARIS) show the localization of 53BP1 (a marker of DSB, red), and PML (green) upon 1-day-long activation of I-PpoI and during recovery from I-PpoI insult. DAPI (blue) and TOTO-3 (white) marked the nucleus and nucleolus, respectively. Only one layer from the several sections is presented. (**D**) The level of DSB upon the recovery from I-PpoI insult was obtained by detecting the 53BP1 (a marker of DSB) by indirect IF andScanR. The histogram represents the frequency of nuclei (%) with the same number of 53BP1 foci. The bin center used for analysis was 2. (**E**) The number of nuclei with PNAs upon the recovery from I-PpoI insult was obtained by detection of PML using indirect IF and a ScanR. The quantification was done manually by evaluating PML localization in more than 200 nuclei; the TOTO-3 was used to mark nucleoli. The bar graph represents the percentage of nuclei with PNAs. Results are presented as a mean ± SEM obtained from four (1A11) and three (1H4) biological replicates. (**F**) The representative images obtained by indirect IF and confocal microscopy (STELLARIS) show the correlation between the localization of B23 (red) in PML-NDS (PML, green) upon the recovery from I-PpoI insult. The nucleus and nucleolus were marked by DAPI (blue), and TOTO-3 (white), respectively. Only one layer from the several sections is presented. (**G**) The representative images obtained by indirect IF and wide-field microscopy show the correlation between the localization of DHX9 (red) and PML-NDS (PML, green) upon the recovery from I-PpoI insult. The nucleus and nucleolus were marked by DAPI (blue), and TOTO-3 (white), respectively. The representative images obtained by indirect IF and confocal microscopy (STELLARIS) show the localization of UBF (**H**, a marker of rDNA; red) or PAF49 (**I**, a subunit of RNAPI; red) in PML-NDS (PML, green) upon the recovery from I-PpoI insult. One layer of the nucleus and three sequential layers of nucleolus with PML-NDS are presented. The nucleus and nucleolus were marked by DAPI (blue) and TOTO-3 (white), respectively. Scale bars, 10 µm (nuclei in **C, F, G, H, and I**) and 5 µm (nucleoli in **H and I**).

The online version of this article includes the following source data and figure supplement(s) for figure 4:

**Source data 1.** Numerical data for D and E.

**Figure supplement 1.** The DNA damage introduced into the rDNA locus by endonuclease I-PpoI induces PML-NDS.

**Figure supplement 2.** The DNA damage introduced into the rDNA locus by endonuclease I-PpoI induces PML-NDS.

**Figure supplement 2—source data 1.** Numerical data for E.

**Figure supplement 3.** The DNA damage introduced into the rDNA locus by endonuclease I-PpoI induces PML-NDS.

**Figure supplement 4.** The DNA damage introduced into the rDNA locus by endonuclease I-PpoI induces cellular senescence.

**Figure supplement 4—source data 1.** Numerical data for A and B.

were absent, consistent with preserved RNAPI activity in most nucleoli (for FU incorporation assay, see *Figure 4—figure supplement 2C*). Note that the rDNA damage-associated 53BP1 foci are linked with both canonical PML-NBs and PML-NDS. However, not all PML-NDS-like structures co-localized or were adjacent to the 53BP1 signal (*Figure 4C* and *Figure 4—figure supplement 2B and D*).

Next, we compared the PML-NDS-like structures induced by I-PpoI with those generated by doxorubicin by examining the accumulation of B23 and DHX9, which are distinct components of doxorubicin-induced PML-NDS (*Imrichova et al., 2019*). Although I-PpoI-induced PML-NDS contained B23 (see *Figure 4F*) with a level higher than that seen in the associated nucleolus (see *Figure 4—figure supplement 2E*), the localization of DHX9 differed from that of doxorubicin-treated cells (*Figure 4G* and *Figure 4—figure supplement 3A and B*), indicating similarities and differences in the composition of both structures. In control cells, DHX9 was homogenously distributed in the nucleus. Two days after doxorubicin treatment, the DHX9 signal was excluded from the nucleolus, and during the recovery period, DHX9 relocalized into the nucleolus and accumulated in PML-NDS (*Figure 4—figure supplement 3A*). In contrast to doxorubicin, two subpopulations of cells emerged after I-PpoI activation: one with DHX9 aggregates inside the nucleolus and the other with a pan-nuclear distribution of DHX9 similar to untreated control cells (*Figure 4G*). The pattern of DHX9 accumulation in PML-NDS, characteristic of doxorubicin response, was present only in a few I-PpoI-induced cells, mainly during the recovery phase (see a gallery of cells in *Figure 4—figure supplement 3B*).

Finally, we examined the localization of RNAPI subunit PAF49 and UBF (activator of pre-rRNA transcription, often used as a marker of rDNA; *Warmerdam et al., 2016*; *Mais et al., 2005*) in

I-PpoI-activated cells. As illustrated in *Figure 4H and I* and *Figure 4—figure supplement 3C and D*, the PML-NDS induced by I-PpoI were located adjacent to the nucleoli, where the UBF and PAF49 signals were detected in cavities distributed uniformly throughout the nucleolus and surrounded by the TOTO-3 signal used as a nucleolar marker. It should be noted that such a localization pattern of UBF and PAF49 indicates the presence of RNAPI activity (see control cells in *Figure 4H and I*). Notably, most of the PML-NDS contained the UBF signal, which was again localized in cavities surrounded by TOTO-3 (*Figure 4H* and *Figure 4—figure supplement 3C*). Similarly, PAF49 showed comparable localization in PML-NDS (*Figure 4I* and *Figure 4—figure supplement 3D*). Since both proteins are markers of rDNA localization, we infer that rDNA is present in at least some PML-NDS.

In our previous work, we demonstrated that the appearance of PNAs following doxorubicin treatment is associated with cell cycle arrest and cellular senescence (*Imrichova et al., 2019*). To understand how selective rDNA damage influences cell fate, we monitored cell viability and colony-forming ability after a transient 24 hr activation of I-PpoI. We observed only a small proportion of cells staining positively for a cell death marker – annexin V (*Figure 4—figure supplement 4A*). Conversely, using a colony formation assay (CFA), we found that only 1% (clone 1H4) and 2% (clone 1A11) of cells, respectively, were able to resume proliferation (*Figure 4—figure supplement 4B and C*). The CFA plate microscopic analysis revealed individual cells exhibiting a senescence-like phenotype interspersed among the colonies (*Figure 4—figure supplement 4D*). To confirm the establishment of senescence, we analyzed senescence-associated β-galactosidase activity 8 and 12 days after the 24 hr activation of I-PpoI. We found that most cells interspersed among the colonies were positive for β-galactosidase (*Figure 4—figure supplement 4E*). These findings suggest that rDNA damage caused by I-PpoI predominantly results in cell cycle arrest and cellular senescence. Therefore, the presence of PNAs may indicate persistent rDNA damage that triggers a long-term senescence-associated cell cycle arrest.

In conclusion, using a model system of enzymatically induced rDNA breaks allowed us to demonstrate that the formation of PNAs directly results from rDNA damage. We have also shown that the composition of I-PpoI-induced PML-NDS resembles the PML-NDS present after more complex DNA damage induced by doxorubicin. Furthermore, the I-PpoI model of rDNA damage helped us to establish that the occurrence of PNAs is associated with a cellular state characterized by persistent rDNA damage, leading to cell cycle arrest and cellular senescence.

## Inhibition of ATM/ATR kinases and RAD51 suppresses the formation of I-PpoI-induced PML-NDS

Our data suggests that direct rDNA damage primarily triggers the formation of the final stage of PNAs, the PML-NDS, which occur as a late response to I-PpoI activation. Previously, we demonstrated through live cell imaging of doxorubicin-treated RPE1*hTERT* cells that PML-NDS originate from funnel-like PNAs, which envelop the nucleolar caps (*Imrichova et al., 2019*). Following I-PpoI activation, both bowl- and funnel-like PNAs, associated with the inactivation of RNAPI activity and rDNA segregation throughout the nucleolus, are absent. It has been shown that the inactivation of RNAPI and the formation of the nucleolar cap following direct rDNA damage induced by I-PpoI activity depend on both ATM and ATR kinase signaling (*Korsholm et al., 2019*; *Mooser et al., 2020*). To investigate whether the formation of PNAs following I-PpoI treatment depends on the activity of these kinases and possibly on the formation of the nucleolar cap, we inhibited ATM or ATR upon activation of I-PpoI (see the experimental scheme in *Figure 5A*). For these experiments, we employed the concentrations of ATM and ATR inhibitors that are commonly used in the DNA damage field (*Zeng et al., 2023*; *Negi and Brown, 2015*; *Golding et al., 2009*; *Ortega-Atienza et al., 2016*; *Zhan et al., 2010*; *Hickson et al., 2004*), as specified in the Materials and methods and Figure legends. The efficiency of ATM and ATR kinase inhibition was confirmed by analyzing the level of the serine 15-phosphorylated p53 after doxorubicin treatment (pS15-p53; *Figure 5—figure supplement 1A*). Subsequently, analyzing the presence of PNAs induced by I-PpoI, we found that the inhibition of ATM, with either of the two commonly used inhibitors (KU-60019, KU-55933) and ATR (with the inhibitor VE-822), respectively, robustly inhibited the formation of PNAs (*Figure 5B*). This finding demonstrates that the emergence of PML-NDS following I-PpoI requires the activity of both these major DDR kinases. Given that both these kinase activities are also essential for the inactivation of RNAPI and the formation of the nucleolar cap (*Korsholm et al., 2019*; *Mooser*

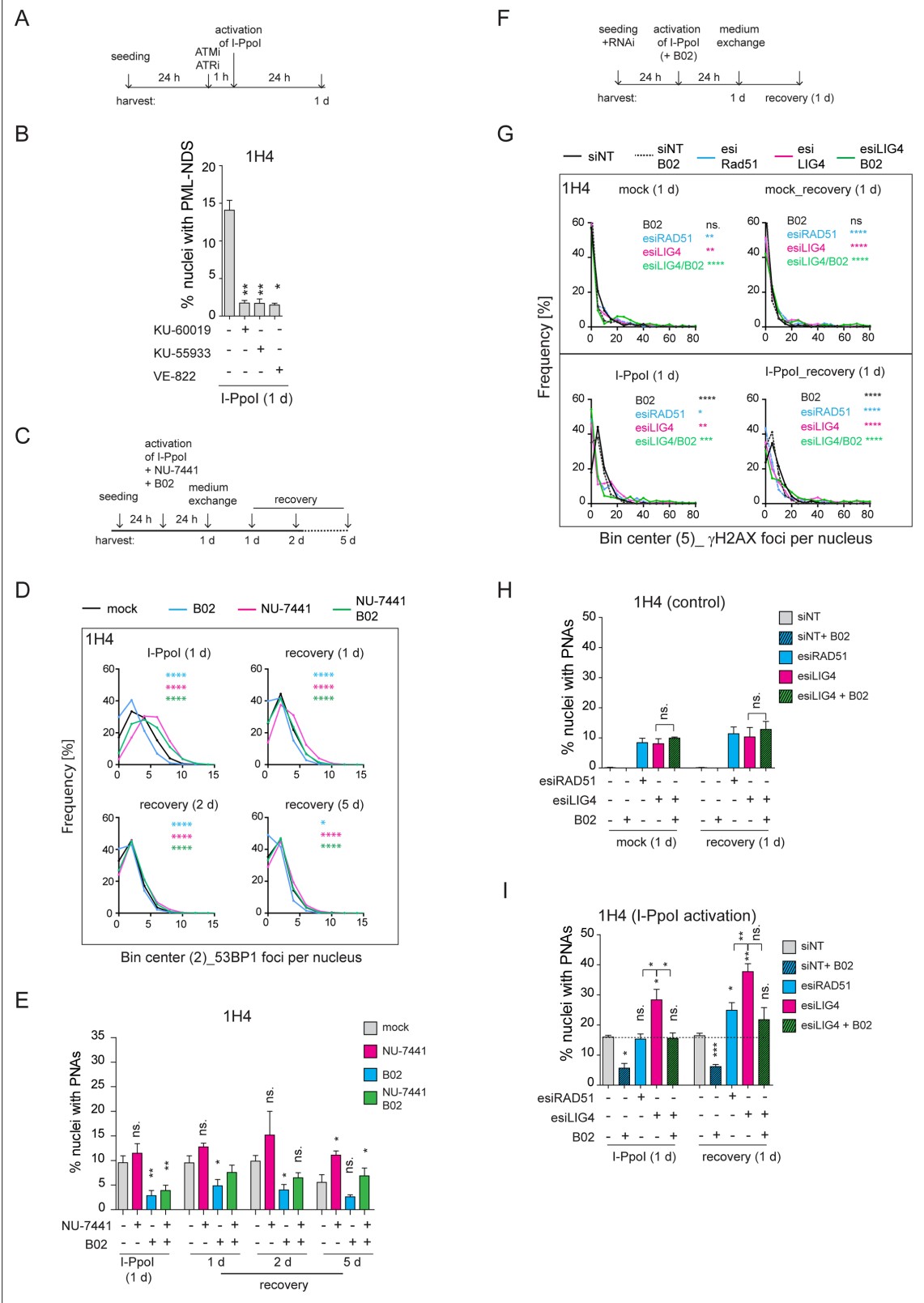

**Figure 5.** Inhibition of ATM, ATR, and RAD51 suppressed the formation of I-PpoI-induced PML-NDS. (**A**) The scheme of the experimental setup used in the experiment shown in (**B**) is presented. Briefly, 24 hr after seeding of RPE-1$^{hTERT}$-I-PpoI cells (clone 1H4), the ATM inhibitors (1 µM KU-60019 or 6 µM KU-55933) or ATR inhibitor (0.2 µM VE-822) were added 1 hr before the activation of I-PpoI expression. (**B**) The number of nuclei with PNAs upon the I-PpoI insult and simultaneous inhibition of ATM or ATR was obtained by detecting PML and nucleolus using indirect IF and ScanR. The quantification

*Figure 5 continued on next page*

*Figure 5 continued*

was done manually, evaluating PML localization in more than 200 nuclei. The bar plot represents the percentage of nuclei with PNAs. (**C**) The scheme of the experimental setup used in (**D and E** and *Figure 5—figure supplement 1B and C*) is presented. Briefly, 24 hr after seeding I-PpoI was activated in RPE-1$^{hTERT}$-I-PpoI cell clones 1A11 and 1H4 for 24 hr. The inhibitors of DNA PK (1 µM NU-7441) or RAD51 (10 µM B02) were applied individually or both together at the time of I-PpoI activation. After 24 hr, the medium was exchanged, and the cells were analyzed during the recovery phase (0, 1, 2, and 5 days). (**D**) During the recovery phase, the level of DSBs was quantified by detecting 53BP1 (a marker of DSBs) using indirect IF and ScanR. The data from three independent biological replicates (values for 200 nuclei) were pooled together and represented as histograms showing the frequency of nuclei (%) with the same number of 53BP1 foci. The bin center used for analysis was 2. (**E**) The number of nuclei with PNAs upon the recovery from I-PpoI insult and simultaneous inhibition of particular DNA damage repair pathway was obtained by detecting PML using indirect IF and ScanR. The quantification was done manually by the evaluation of PML localization in more than 200 nuclei. (**F**) The scheme of the experimental setup used in experiments shown in (**G, H, and I**) is presented. Briefly, the RPE-1$^{hTERT}$-I-PpoI (clone 1H4) were transfected by interfering RNA upon seeding. After 24 hr, the I-PpoI was activated for 24 hr. Then, the medium was changed, and cells recovered from rDNA damage for 0 and 1 day. The control cells were treated with corresponding concentration of DMSO simultaneously (mock). (**G**). The level of DSB was quantified by detecting γH2AX (a marker of DSBs) using indirect IF and ScanR. The data from three independent biological replicates (values for 200 nuclei) were pooled together and represented as histograms showing the frequency of nuclei (%) with the same number of γH2AX foci. The number of nuclei with PNAs upon the recovery from (**H**) inhibition of a particular DNA damage repair pathway by RNA interference and without the activation of I-PpoI or (**I**) when I-PpoI was activated together with the inhibition of a particular DNA damage repair pathway was obtained by detection of PML using indirect IF and ScanR. The quantification was done manually by evaluating PML localization in more than 200 nuclei. The bar plot represents the percentage of nuclei with PNAs. Results are presented as a mean ± SEM obtained from at least three biological replicates. A paired (**B, E, H, and I**) and unpaired (**D, G**) two-tailed t-test were used for statistical evaluation. Asterisks indicate the following: ****p<0.0001, ***p<0.001, **p<0.01, *p<0.05.

The online version of this article includes the following source data and figure supplement(s) for figure 5:

**Source data 1.** Numerical data for B, D, E, G, H, and I.

**Figure supplement 1.** Inhibition of ATM, ATR, and RAD51 suppressed the formation of I-PpoI-induced PML-NDS.

**Figure supplement 1—source data 1.** RAW membranes for A and D.

**Figure supplement 1—source data 2.** Uncropped and labeled membranes for A and D.

**Figure supplement 1—source data 3.** Numerical data for A, B, C, D, and E.

**Figure supplement 2.** RAD51 suppressed the formation of I-PpoI-induced PML-NDS.

**Figure supplement 2—source data 1.** Numerical data for C and D.

---

*et al., 2020*), we hypothesized that the presence of a nucleolar cap is critical for the emergence of PML-NDS.

As the nucleolar caps may also provide an interface for HDR (*van Sluis and McStay, 2015*; *Harding et al., 2015*), we next explored the impact of experimentally altered DNA repair on the formation of I-PpoI-induced PML-NDS. Thus, we activated I-PpoI and concurrently applied chemical inhibitors of HDR (B02) and NHEJ (NU-7441), individually or in combination, to inhibit these major DSB repair pathways (see the experimental scheme in *Figure 5C*). We then assessed the extent of (r)DNA damage (53BP1 foci per nucleus) and the number of nuclei with PNAs using indirect IF and high-content microscopy. These experiments were conducted in two cell clones expressing I-PpoI, 1H4, and 1A11, 24 hr post-I-PpoI activation and during the recovery phase (1, 2, and 5 days after medium exchange/I--PpoI deactivation). As depicted in the histograms in *Figure 5D* (clone 1H4) and *Figure 5—figure supplement 1B* (clone 1A11), inhibiting NHEJ with NU-7441 during I-PpoI activation significantly increased the number of (r)DNA damage foci at all time-points compared to the control treatment. In contrast, inhibiting HR with B02 resulted in fewer 53BP1 foci than in control cells. Combining both drugs together did not further increase the number of 53BP1 foci compared to DNA PK inhibition alone, but rather, the resulting extent of I-PpoI-induced (r)DNA damage was lower. These findings are consistent with the proposed notion that NHEJ is the primary pathway involved in the repair of I-PpoI-induced DSBs and indicate that when HDR is blocked, the I-PpoI-induced rDNA damage is repaired more efficiently (*Warmerdam et al., 2016*).

In terms of the impact of DNA repair inhibition on the formation of I-PpoI-induced PML-NDS, we found that inhibiting NHEJ with NU-7441 led to an increased number of cells with PML-NDS. This was in line with a higher incidence of DSBs. However, this trend was only significant at a one time point (1H4; 5 days recovery; see *Figure 5E* and *Figure 5—figure supplement 1C*). Conversely, inhibiting the formation of RAD51 filaments with B02 significantly reduced the number of cells with PML-NDS in both clones 24 hr post-treatment, consistent with decreased DNA damage markers. During the recovery phase, the number of cells with PML-NDS following B02 treatment remained lower, with

this decrease being statistically significant at most time points. Interestingly, the concurrent application of both inhibitors led to some intriguing results. Although inhibiting both primary repair pathways increased (r)DNA damage (albeit not as extensively as in cells treated solely with NU-7441), the number of cells with PML-NDS was significantly lower 24 hr post-I-PpoI activation compared to cells without inhibitor application. It is important to note that this significant difference leveled out during the recovery phase following the washout of the DNA repair pathway inhibitors. Furthermore, 5 days post-recovery from the I-PpoI insult, the number of cells with PML-NDS was comparable to the control population (I-PpoI only). These findings suggest that the addition of B02, which blocks the formation of RAD51 filaments, inhibits the pathway critical for the formation of PML-NDS.

To further demonstrate that a defect in RAD51 filamentation impairs the formation of PNAs following I-PpoI activation, we downregulated the RAD51 protein using a pool of siRNA targeting RAD51 mRNA (esiRAD51). Furthermore, to mimic the effect of DNA PK inhibition on NHEJ, we downregulated DNA ligase IV (LIG4), an essential component of this repair pathway (*Grawunder et al., 1998*). Again, for the latter knockdown (KD), a pool of siRNA targeting LIG4 mRNA (esiLIG4) was used. The efficiency of the KD was confirmed by western blotting and RT qPCR (*Figure 5—figure supplement 1D and E*). In silenced cells, we tracked DSBs and PML localization 24 hr post-I-PpoI activation and 24 hr post-recovery from direct rDNA damage (see the scheme in *Figure 5F*). As shown in the histogram and image galleries (*Figure 5G* and *Figure 5—figure supplement 2A and B*), the patterns of γH2AX foci differed between the control cells (siNT) and the cells where LIG4 or RAD51 expression was suppressed. Notably, in cells with either RAD51 or LIG4 KD, even without the activation of I-PpoI, a significantly higher number of cells harbored more than 35 γH2AX foci per nucleus, indicating that the downregulation of either RAD51 or LIG4 alone resulted in enhanced DSBs (see graphs in *Figure 5—figure supplement 2C*). Furthermore, after 24 hr I-PpoI activation in cells deficient for RAD51, LIG4, or treated with B02, there was an increased population of cells with almost no γH2AX signal around the nucleolus (*Figure 5G* and *Figure 5—figure supplement 2A, B and D*). This difference suggests that following RAD51 KD, LIG4 KD, or B02 inhibitor addition, the dynamic of rDNA repair was altered compared to the control population (only activation of I-PpoI). Thus, the cells with inhibited NHEJ or HDR exhibit both the I-PpoI-induced rDNA DSBs (modulated by RAD51/LIG4 deficiency) as well as additional I-PpoI-independent DNA damage (likely also in rDNA) reflecting the deficiency of RAD51 and LIG4 and hence the ensuing inability to cope efficiently with the omnipresent endogenous DNA damage.

Consistently with the observed extent of baseline DSB, we found that the KD of RAD51 and LIG4 (+/-B02), even without the I-PpoI activation, induced PML-NDS at both tested time points (*Figure 5H*). Moreover, there was no significant difference in the number of nuclei with PML-NDS between RAD51 and LIG4 KD under baseline conditions. This scenario changed when I-PpoI was expressed (*Figure 5I*). Simultaneous ablation of RAD51 and 24-hour-long I-PpoI activation did not increase the number of nuclei with I-PpoI-induced PNAs compared to the control. Conversely, the KD of LIG4 resulted in a higher number of nuclei with I-PpoI-induced PNAs than the population of cells treated with control siRNA or esiRAD51. Combining B02 and esiLIG4 upon I-PpoI activation (simultaneous impairment of NHEJ and HDR) significantly reduced the number of cells with PNAs. Interestingly, such reduction of PNAs after B02 co-treatment was not observed upon the ablation of LIG4 activity in control cells (lacking I-PpoI induction), indicating that LIG4 depletion alone leads to accumulation of a different type of rDNA (endogenous) damage than the rDNA DSBs triggered by the endonuclease I-PpoI and that in response to such endogenous rDNA damage, the B02 co-treatment did not attenuate the signal towards PNAs.

Furthermore, we tracked PML localization after a 24 hr recovery from rDNA damage (*Figure 5I*). At this time point, RAD51 knockdown induced a significantly higher number of nuclei with PNAs compared to control cells. However, this number was still significantly lower than the number of PNAs-positive nuclei after esiLIG4, and comparable to that after combined esiLIG4 and B02 treatment. Thus, although RAD51 knockdown did not reduce the number of nuclei with PNAs as robustly as seen after B02 treatment, the number of cells with PNAs was lower than after LIG4 knockdown and comparable with cells in which LIG4 knockdown was combined with B02.

In summary, we found that after inducing DSBs in rDNA by I-PpoI, the activities of both ATM and ATR kinases, which are vital for RNAPI inhibition and nucleolar cap formation, were essential for the emergence of PNAs. Additionally, the modulation of DNA repair pathways of I-PpoI-induced DSBs

revealed that cells with defects in NHEJ were more susceptible to PNAs formation than cells with non-functional RAD51. Moreover, through these experiments, we found that the sole deficiency of RAD51 or LIG4 can also trigger the signal for PNAs formation, indicating that the development of this PML structure is linked to not only persistent I-PpoI-induced DSBs but also as a consequence of unrepaired endogenous rDNA damage.

## I-PpoI-induced PML-NDS co-localize with resected DNA and are present mainly in the G1 cell cycle phase

The results of our present study so far link the occurrence of PNAs to the formation of the nucleolar cap with persistent rDNA damage and the interplay with DNA repair. Specifically, as shown above, the PNAs contain markers of DSBs, such as 53BP1 and γH2AX (*Figure 4C* and *Figure 4—figure supplements 1 and 2A and B*) and RAD51 deficiency impairs the occurrence of PNAs following an rDNA lesion introduced by I-PpoI. To better elucidate the relationship between PNAs and HDR, we analyzed the presence of the RAD51 foci (indicating HDR activity), and the extent of DSB end resection marked by the phosphorylated form of RPA32 (pS33; pRPA) following I-PpoI activation. As expected, we found that RAD51 foci were primarily present in S/G2 cells (estimated according to the DNA content), and their number gradually decreased with prolonged induction of I-PpoI (6, 8, and 24 hr; *Figure 6A*). After 24 hr of I-PpoI induction, a time point at which PNAs start to emerge, RAD51 foci were only observed in a few cells (see *Figure 6—figure supplement 1A*). Importantly, the nuclei containing PNAs lacked any detectable RAD51 signal. Note that the RAD51-positive nuclei are shown as a control of RAD51 staining (see *Figure 6B*).

We then analyzed the localization of pRPA. Notably, the pRPA foci were present in the nuclei of cells in both G1 and S/G2 cell cycle phases, and their number was highest after 24 hr of I-PpoI activation, a time point at which PML-NDS are already present (*Figure 6C* and *Figure 6—figure supplement 1B*). Using confocal microscopy, we found that most PNAs are associated with the pRPA signal (*Figure 6D*). These results suggest that PNAs formed after direct rDNA damage are not linked to activation of the ongoing canonical HDR, as PNAs emerge at the time points when RAD51 expression is attenuated. As PNAs associate with resected DNA (pRPA) and markers of DSBs (53BP1 and γH2AX), yet RAD51 is lacking, we assume that the persistent rDNA lesions initially directed towards the nuclear cap-associated HDR can signal for PNAs.

Finally, to provide a broader perspective on cell proliferation, we investigated whether PNAs are preferentially formed during any specific cell cycle phase(s). Utilizing high-content image acquisition and analysis, we categorized cells into G1 and S/G2 populations based on total DAPI fluorescence intensity as a readout of DNA content (see *Figure 6—figure supplement 1C and D*). From the obtained image galleries, we calculated the percentage of nuclei with PNAs present in the respective cell cycle phases (see *Figure 6E*). After 24 hr of I-PpoI activation, PNAs were present in both G1 and S/G2 cells, however, the G1 cells contained a significantly higher proportion of nuclei with PNAs. Upon the recovery phase, the number of nuclei with PNAs was equal in both populations. We also analyzed the distribution of PNAs in relation to cell cycle position after RAD51 or LIG4 KD. We found that after depletion of LIG4 or RAD51 combined with 24 hr activation of I-PpoI, the PNAs-positive nuclei were again present in both G1 and S/G2 cells, yet their percentage was significantly higher in S/G2. Upon the recovery phase, the significantly higher occurrence of nuclei with PNAs in S/G2 was found only after the LIG4 KD. The same analysis in cells in which only RAD51 or LIG4 was ablated (i.e. without I-PpoI activation) showed that the PNAs occurred significantly more frequently in S/G2 at both time points (*Figure 6F*). Note that *Figure 6—figure supplement 1E* presents a different plot of the same results, better illustrating that at 24 hr post-I-PpoI activation, the cells with impaired RAD51 (by B02 or siRNA) featured decreased PNAs in G1. In S/G2 cells, the B02-mediated RAD51 inhibition did not affect the percentage of nuclei with PNAs, in contrast to esiRAD51-mediated depletion of RAD51 protein that caused a significant increase of PNA-positive cells, indicating a differential impact of RAD51 functional inhibition *versus* the absence of the RAD51 protein. All these results indicate that although nuclei with PNAs can be present in G1 or S/G2 cells, the specific treatment (type of rDNA damage) can affect the probability of their emergence in particular cell cycle phases. After rDNA damage introduced by I-PpoI, PNAs tend to occur mainly in G1, and RAD51 ablation attenuates their emergence. Importantly, the sole downregulation of either LIG4 or RAD51 (without I-PpoI induction) leads to a kind of rDNA damage that shifts PNAs occurrence primarily to the S/G2 cell. For

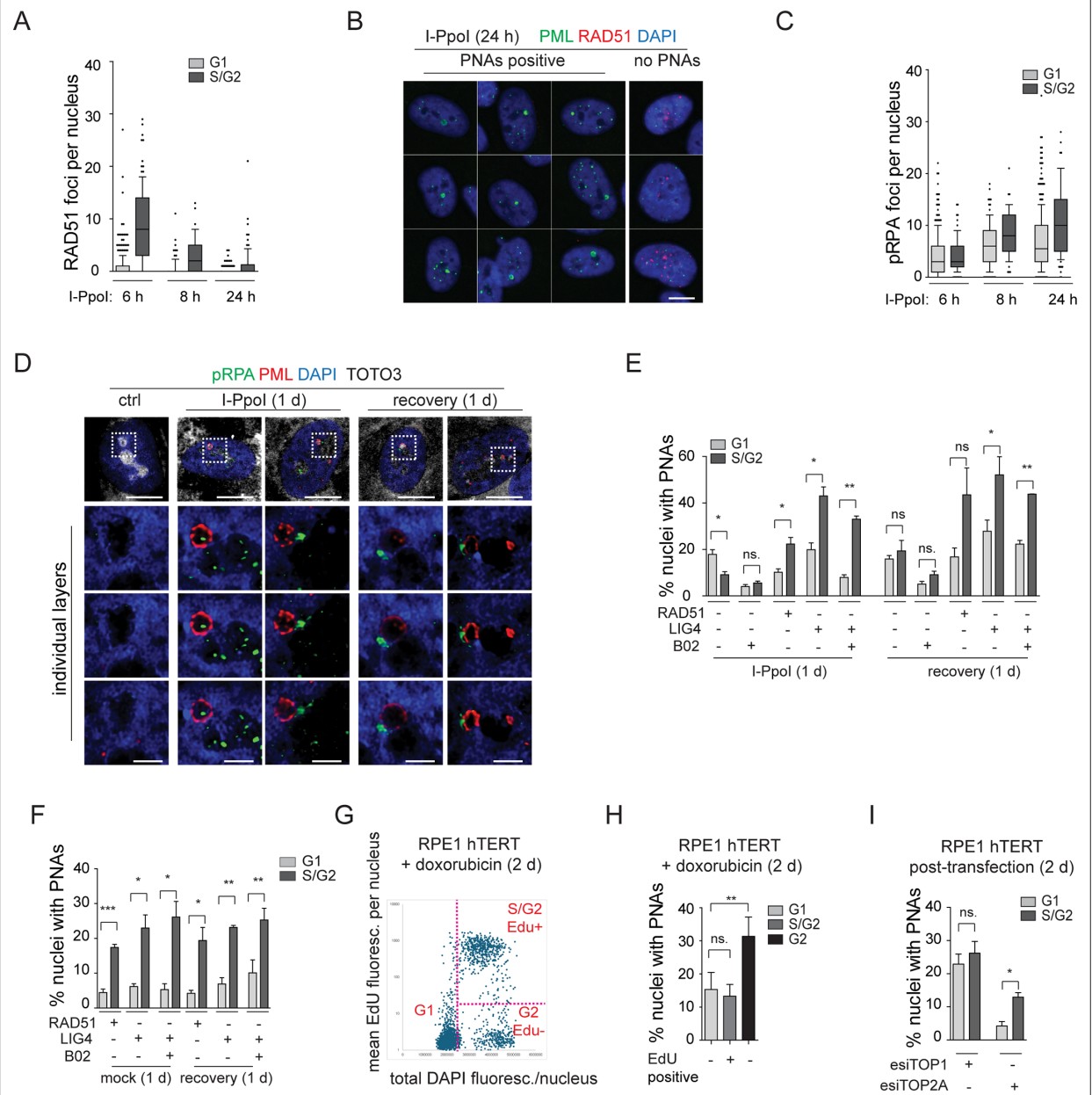

**Figure 6.** The resected DNA is present in both G1 and S/G2 cells and co-localizes with I-PpoI-induced PNAs. (**A**) The I-PpoI expression was activated 24 hr after RPE-1$^{hTERT}$-I-PpoI (1H4) seeding. The cells were harvested 6, 8, and 24 hr after activation of I-PpoI expression, and RAD51 and PML were detected by indirect IF. The nuclei were stained with DAPI. The number of RAD51 foci and total DAPI fluorescence in individual nuclei were evaluated using the ScanR software. The G1 and S/G2 cells were estimated according to the total DAPI fluorescence. The count of RAD51 per nucleus is shown for G1 and S/G2 cells using the whiskers plot (box shows 10–90 percentiles; the black line is median). (**B**) The cells harvested after 24 hr of I-PpoI activation were stained for PML (green) RAD51 (red) and nucleus (blue). The characteristic nuclei with PNAs are shown. (**C**) The experimental setup described in (**A**) was used, but RPA2pS33 and PML were detected by indirect IF and ScanR. The number of RPA2pS33 foci and total DAPI fluorescence in individual nuclei were evaluated using the ScanR software. The G1 and S/G2 cells were estimated according to the total DAPI fluorescence. The count of RPA2pS33 per nucleus is shown for G1 and S/G2 cells using the whiskers plot (box shows 10–90 percentiles; the black line is median). (**D**) PML (red) and resected DNA (RPA2pS33; green) were detected after 24 hr long activation of I-PpoI and after 1 day-long recovery using indirect IF and confocal microscopy (STELLARIS). The whole nucleus and three separate layers of nucleolus are shown. Nuclei were stained by DAPI (blue) and nucleoli by TOTO3 (white). Scale bar, 10 µm (nucleus); 2 µm (nucleolus). (**E**) RPE-1$^{hTERT}$-I-PpoI (1H4) were transfected by interfering RNA upon seeding. After 24 hr, the I-PpoI was activated by doxycycline and Shield for another 24 hr. Then, the medium was changed, and cells recovered for 1 day. The PML was detected by indirect IF and ScanR. ScanR analysis software divided the cells according to total DAPI fluorescence as a G1 and S/G2. Then, the gallery of nuclei was made for each group, and the number of nuclei with PNAs was manually calculated. The percentage of nuclei with PNAs presented in G1 and S/G2 is shown as a column graph. (**F**) The experimental setup and evaluation were like those in (**E**). Only the I-PpoI was not activated; instead, DMSO

*Figure 6 continued on next page*

*Figure 6 continued*

was added. (**G**) RPE-1$^{hTERT}$ cells were treated with 0.75 μM doxorubicin and EdU for 48 hr. Then, the PML was detected by indirect IF and EdU by ClickIt chemistry. ScanR analysis software divided the cells according to total DAPI fluorescence and EdU fluorescence into three groups: G1 (Edu-), S/G2 (Edu+), and G2 (EdU-). The scatter plot shows the total DAPI fluorescence and EdU mean fluorescence of individual nuclei and gates for the mentioned three groups. (**H**) Galleries of nuclei from gates shown in (**G**) were made, and the number of nuclei with PNAs was manually calculated. The percentage of nuclei with PNAs presented in G1(EdU-), S/G2(EdU+), and G2(EdU-) is shown as a column graph. (**I**) RPE-1$^{hTERT}$were transfected by interfering RNA targeting TOP2A and TOP1 upon seeding. After 2 days, PML was detected by indirect IF and ScanR. ScanR analysis software divided the cells according to total DAPI fluorescence as a G1 and S/G2. Then, the galleries of nuclei were made, and the number of nuclei with PNAs was manually calculated. The percentage of nuclei with PNAs presented in G1 and S/G2 is shown as a column graph. Scale bar, 10 μm (nuclei in **B, D**); 2 μm (nucleoli in **D**). Results are presented as a mean ± SEM obtained from at least three biological replicates. A paired two-tailed t-test was used for statistical evaluation. Asterisks indicate the following: ****$p<0.0001$, ***$p<0.001$, **$p<0.01$, *$p<0.05$.

The online version of this article includes the following source data and figure supplement(s) for figure 6:

**Source data 1.** Numerical data for A, C, E, F, G, H, and I.

**Figure supplement 1.** The resected DNA is present in G1 and S/G2 cells and co-localizes with I-PpoI-induced PNAs.

**Figure supplement 1—source data 1.** Numerical data for C and D.

comparison, we conducted a similar cell cycle analysis following doxorubicin treatment. It is important to note that EdU was added concurrently with doxorubicin to detect cells undergoing DNA replication. Two days after doxorubicin addition, PML, and EdU were detected using high-content image acquisition and analysis. The cells were categorized into three groups based on the EdU signal and total DAPI fluorescence intensity, as follows: (i) G1, (ii) S/G2 EdU-positive, and (iii) G2 EdU-negative (see *Figure 6G*). We then calculated the percentage of nuclei with PNAs in each group. As shown in *Figure 6H*, PNAs were present in all groups, but the highest percentage was observed among the G2 EdU-negative cells. This result suggests that cells that were already in G2 (with sister chromatids held together), upon adding doxorubicin, incurred such (r)DNA damage that increased the likelihood of PNAs induction. Interestingly, the sole depletion of TOP2A preferentially introduced PNAs in the S/G2 cells, while after the depletion of TOP1, the distribution of PNAs between the G1 and S/G2 phases was equal (*Figure 6I*). This observation further indicates that the signal for PNAs can arise from different types of damage introduced into the rDNA locus.

In summary, we demonstrated that the I-PpoI-induced PNAs formed at a time point when RAD51 is attenuated, yet the PNAs still associated with markers of DSBs and DNA resection. This pattern indicates that the signal for PNAs arises from persistent DNA lesions initially directed for HDR. Finally, we found that PNAs can be present in G1-S-G2 cell cycle phases. However, the nature of the genotoxic insult strongly influences the probability of PNAs formation in relation to cell cycle phase position.

In conclusion (for the suggested model, see the scheme in *Figure 7*), our data indicate that the PNAs result from: (1) persistent rDNA DSB lesions, which were initially directed for HDR and have already been resected; and (2) topological aberrations in nucleolar caps/NORs. Overall, as discussed in more detail below, our study suggests that PNAs can potentially segregate such rDNA damage loci away from the active nucleolus.

## Discussion

While the phenomenon of PML relocation to the nucleolus or its vicinity under various treatments has been known for years (*Janderová-Rossmeislová et al., 2007*; *Bernardi et al., 2004*; *Condemine et al., 2007*), the primary signal driving the interaction of PML with the nucleolus has remained unclear. Our present study sheds light on this area of cell (patho)biology, by providing evidence that persistent rDNA damage, combined with topological stress and inhibition of RNAPI are key characteristics of the most potent inducers of PML-nucleolar associations (PNAs).

Doxorubicin, a TOP2 inhibitor and one of the most potent inducers of PNAs, triggers DSBs within the rDNA/DJ/NOR locus, followed by a subset of such DSBs aggregating into the nucleolar cap. This aggregated structure not only co-localizes with bowl- and funnel-like PNAs but also with PML-NDS formed after the recovery of RNAPI activity. This suggests a potential mechanism for excluding damaged rDNA/DJ/NORs from the active nucleolus.

The connection between rDNA damage and signaling that triggers PNAs formation is further strengthened by the observation that direct rDNA damage, induced by the endonuclease I-PpoI,

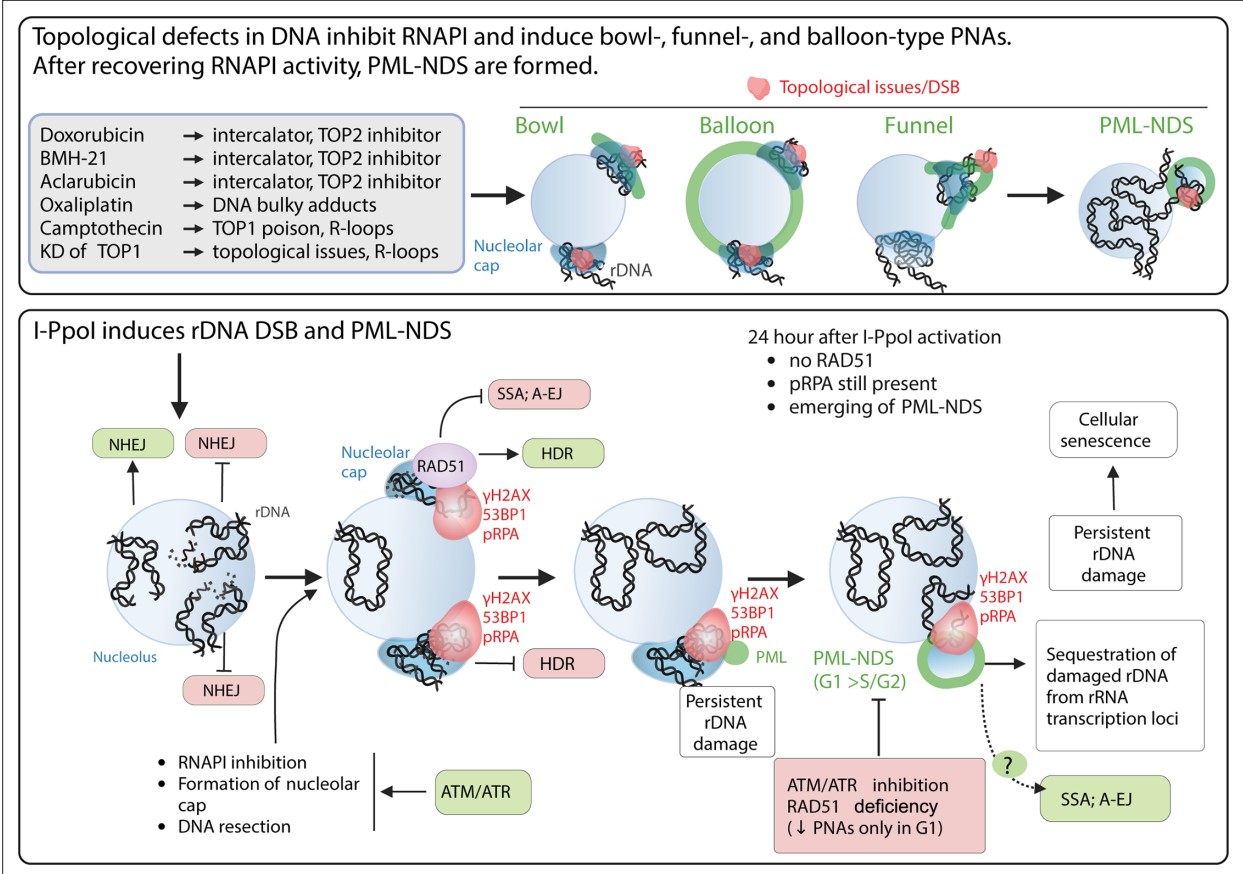

**Figure 7.** The schematic presentation of signals triggering the development of PNAs. The *upper* panel represents the most potent PNA inductors. The common characteristic of these treatments is the ability to alter chromatin topology and simultaneously cause the 'long-term inhibition' of RNAPI (the inhibition was present even after 2 d long treatment). After these treatments, the bowl-, funnel-, and balloon-like PNA types interacting with nucleolar caps are formed. After recovery of RNAPI, the persistent rDNA damage is sequestrated, resulting in the formation of PML-NDS. The *lower* panel presents the signals affecting the PNAs formation after I-PpoI-induced rDNA DSBs. Such rDNA DSBs are predominantly repaired by NHEJ in the nucleolar interior (***Warmerdam et al., 2016***; ***Harding et al., 2015***). Unrepaired rDNA lesions signal for ATM/ATR-dependent inactivation of RNA polymerase I and the translocation of damaged rDNA to the nucleolar periphery, forming a nucleolar cap (***Korsholm et al., 2019***; ***Harding et al., 2015***). The damaged rDNA in the nucleolar cap is resected, suggesting repair by HDR (***Warmerdam et al., 2016***; ***van Sluis and McStay, 2015***). Our data indicate that I-PpoI-induced PNAs (PML-NDS) form as a late response to rDNA damage, occurring when RAD51 foci in the nucleolar periphery decline. Markers of DSBs (γH2AX, 53BP1) and DNA resection (pRPA) still co-localize with PNAs, supporting the hypothesis that persistent rDNA DSBs are incompatible with NHEJ or HDR signal for PNA development. The exact functions of PNAs are unknown, but we hypothesize that they isolate damaged rDNA from active pre-rRNA transcription foci. We cannot exclude a link to alternative repair pathways for persistent rDNA damage. Notably, I-PpoI-induced PML-NDS are mainly present in the G1 cell cycle phase. Inhibiting ATM/ATR, which is essential for RNAPI inhibition, nucleolar cap establishment, and HDR, prevents their formation. RAD51 ablation negatively affects the occurrence of I-PpoI-induced PNAs. Importantly, the PNAs decline was present only in G1. The mechanisms by which RAD51 affects PNA establishment are unclear, but RAD51 promotes HR and inhibits alternative pathways such as SSA and A-EJ or regulates DNA end processing. The scheme was created using BioRender.com and published using a CC BY-NC-ND license with permission (Agreement number: GD279KWD6V).

shares the ability to induce PNAs. Moreover, considering that I-PpoI-induced PNAs appear when the expression of RAD51 is reduced, yet these structures co-localize with resected DNA, we propose that the underlying molecular mechanisms are tied to the generation and processing of a specific, persistent (difficult-to-repair) type of rDNA topological alterations or DNA lesions.

## PNAs indicate the persistent rDNA damage that can trigger cellular senescence

We propose that persistent rDNA lesions, challenging to repair and eliciting combined ATM- and ATR-mediated kinase activity, provide the so-far elusive signal required for the interaction of PML with nucleolar caps and the formation of PNAs. Persistent unresolved DNA lesions are believed to be

important inducers of cellular senescence, yet the molecular differences between transient (repairable), persistent, and permanent DNA lesions are not well defined.

DSBs are mostly resolved within the first 24 hr (*Rodier et al., 2011*; *Martinez-Pastor et al., 2021*). Yet, even beyond this time point, both the percentage of DSB-positive nuclei and the number of DSBs per nucleus can decrease (*Fumagalli et al., 2012*; *Rodier et al., 2009*). This suggests that even some of the more persistent DNA lesions have the potential to be repaired.

In the context of rDNA DNA damage response (DDR) field, the term 'persistent lesions' is used to describe rDNA lesions that signal for ATM-dependent inhibition of RNAPI, followed by the aggregation of the damaged rDNA locus in the nucleolar cap (*Korsholm et al., 2020*; *Harding et al., 2015*). The primary attribute of these lesions is that they cannot be quickly repaired by NHEJ (that operates inside the nucleolus) and require additional processing for resolution. Such lesions begin to accumulate up to 3 hr after direct rDNA damage, although this timing can vary depending on the model used (*Korsholm et al., 2019*; *Mooser et al., 2020*; *Harding et al., 2015*).

Interestingly, we observed that I-PpoI-induced PNAs begin to appear in an increased number of nuclei 24 hr after I-PpoI activation, coinciding with the recovery of RNAPI activity in most nucleoli. Notably, I-PpoI-induced PNAs (mostly PML-NDS) associate with rDNA DSBs, which contain resected DNA but lack RAD51. Therefore, these rDNA lesions cannot be repaired by either NHEJ or HDR, and if they are resolved, an alternative DDR pathway is likely involved.

As previously mentioned, persistent DSBs signal for cell cycle arrest that, if prolonged, can lead to cellular senescence. We have previously demonstrated that doxorubicin treatment, which induces PNAs, also triggers cellular senescence (*Imrichova et al., 2019*). Our present study shows that after direct rDNA damage, which in parallel induces PNAs, most cells enter a senescent state. Both these observations suggest that the formation of PNAs indicates the presence of persistent (difficult-to-repair) rDNA damage, the signaling which may trigger induction of cellular senescence.

However, the impact of PNAs on the resolution of such persistent rDNA lesions needs to be determined. Importantly, PML-NDS were observed during the replicative senescence of hMSC (*Janderová-Rossmeislová et al., 2007*), suggesting that unresolved issues in the rDNA locus are linked to both acute (such as drug- or I-PpoI-induced rDNA damage), as well as population doubling limit-associated DNA damage accumulating with cellular senescence.

## The persistent DSBs directed for HDR are the interface for PNAs triggering

Our data strongly suggests a complex interplay between the presence of rDNA damage and stimuli that trigger PNAs. This is not only due to the observation that genotoxic treatments induce PML interaction with the nucleolar cap but can also be supported by direct evidence. After treatment with doxorubicin, we found that most PNAs interacted with rDNA/DJ, which tested positive for markers of DSBs. Additionally, the DSB introduced directly into the rDNA locus by endonuclease I-PpoI triggered the PML-nucleolar association.

The relationship between rDNA damage and the formation of PNAs was further supported by experiments in which DNA repair pathways were ablated. These experiments not only demonstrated a link between the signal for PNAs formation and the abortion of DNA repair but also suggested that rDNA DSBs directed for HDR induce the PML-nucleolar association.

Our present study shows that the concurrent inhibition of RAD51 filamentation and doxorubicin treatment significantly increased the number of nuclei with PNAs, whereas blocking NHEJ by DNA PK inhibitor did not significantly affect the frequency of PNAs. This finding suggests that when HDR is restricted, there is a higher level of unresolved or persistent damage in the rDNA/DJ locus, thereby amplifying the signal leading to PNAs. According to the literature, the repair of DNA damage induced by doxorubicin relies on both major DSB repair pathways, NHEJ and HDR (*Olivieri et al., 2020*). However, it has also been demonstrated that DNA repair of doxorubicin-induced DNA damage depends less on NHEJ in comparison with, e.g., etoposide-induced DNA damage (*Maede et al., 2014*) and that the former repair is prolonged (*Pang et al., 2013*). This supports the hypothesis that the doxorubicin treatment induces more complex DNA damage. Notably, the direct involvement of RAD51 (and HDR) in the repair of DNA damage induced by doxorubicin was observed in multiple myeloma and HCT-116 colon carcinoma cells. In both models, the combination of B02 and doxorubicin resulted in an increased number of DSBs and apoptosis (*Alagpulinsa et al., 2014*; *Schürmann*

et al., 2021). Therefore, we can infer that after exposure to doxorubicin, the inhibition of HDR rather than inhibition of NHEJ leads to unresolved and persistent damage in the rDNA locus, which triggers the formation of PNAs.

The second observation showed that TDP2 ablation upon etoposide treatment significantly elevated PNAs. Considering that the TDP2-dependent removal of TOP2 adducts after etoposide treatment is essential for successful DNA repair (Cortes Ledesma et al., 2009; Canela et al., 2019), this again supports the notion that persistent DNA damage amplifies the signal for PNAs.

The experimental inhibition of DNA repair pathways, together with direct rDNA damage induced by I-PpoI, resulted in some unexpected findings. In all experimental setups, the restriction of NHEJ (by DNA PKi or LIG4 KD) induced more nuclei with PNAs than the restriction of HDR (by RAD51 ablation). However, when the NHEJ-deficient cells were compared with control cells after I-PpoI-induced rDNA damage, the significant elevation of nuclei with PNAs was observed only after the downregulation of ligase IV (esiLIG4). Interestingly, when HDR was inhibited by B02, and simultaneously, the NHEJ repair pathway was blocked by either NU-7441 or esiLIG4, the number of nuclei with PNAs significantly declined, suggesting a dominant effect of HDR inhibition in such setting.

These results imply that when NHEJ alone is inhibited, the I-PpoI-induced DSBs are more likely to trigger PNAs formation. However, the restriction of HDR (RAD51 ablation), even in the absence of functional NHEJ, attenuates PNAs formation. The latter observation raises the question of why defects in HDR block PNA formation despite the non-functionality of both major repair pathways.

Our additional insights in this context imply that directing rDNA repair for HDR is an essential step for PNAs formation. First, we observed a significant decline in the number of nuclei with PNAs when we inhibited ATM or ATR kinase signaling. Several groups reported that the activity of both kinases is essential for the inhibition of RNAPI and the formation of the nucleolar cap, which is the interface for HDR of rDNA DSB unrepaired by NHEJ (Korsholm et al., 2019; Mooser et al., 2020; Canela et al., 2019).

Furthermore, the occurrence of PNAs during the cell cycle showed that after 24 hr I-PpoI activation, the G1 cells contained a higher proportion of nuclei with PNAs than the S/G2 cell population. Our concept that PNAs can be triggered by persistent rDNA damage implies that the I-PpoI-induced rDNA lesions are more efficiently repaired in S/G2, where NHEJ and HDR work in parallel. Additionally, when RAD51 function was eliminated, the PNAs frequency was lower only in G1 compared to the control. This indicates that eliminating the RAD51 function reduces the signal for PNAs in G1, in the cell cycle phase where HDR should be inhibited, and the resection of DNA tightly regulated. Though speculative, this might indicate that some residual HDR (RAD51) activity operates even in G1, and when it is ablated, the PNAs formation is attenuated. In line with this, van Sluis et al. showed that HDR components are indeed present in nucleolar caps even in G1 phase in response to direct rDNA damage, indicating that HDR is not fully restricted in G1 and can be at least partially active (van Sluis and McStay, 2015).

The principal observation we gained was that PNAs are formed at the time point when RAD51 expression is attenuated. However, PNAs interact with the pRPA, which marks DNA resection. Thus, we can assume that G1 rDNA lesions incompatible with NHEJ accumulate in the nucleolar cap for repair. These lesions are resected, but the recombination event has probably not been completed. Such incompletely processed lesions can signal the formation of PNAs. Thus, eliminating RAD51 can negatively affect the development of some HDR intermediates important for the establishment of PNAs. Interestingly, it was recently published that RAD51, by binding ssDNA, inhibits alternative repair pathways such as single-strand annealing (SSA) or alternative end-joining (A-EJ). Additionally, the depletion of RAD51 promotes repair not by NHEJ but by these two backup repair mechanisms (So et al., 2022). Considering this, our data may indicate that the ablation of the RAD51 function activates SSA/A-EJ-mediated repair of the subset of rDNA lesions present in G1 cells that are incompatible by NHEJ and cannot be solved by HDR either, as the sister chromatid template for HR is unavailable.

It is important to note that the existence of PNAs is not restricted to any specific phase of the cell cycle, as we observed PNAs in the G1, S, and G2 phases. However, the type of treatment applied significantly influences the likelihood of PNA formation in a particular cell cycle phase.

For instance, TOP1 knockdown triggers PML nuclear interaction in both G1 and S/G2 phases. This is because TOP1 is involved in the removal of both negative and positive DNA supercoiling during DNA replication and transcription (Pommier et al., 2016), indicating that its activity is not

cell cycle-dependent. In contrast, the knockdown of TOP2A primarily results in PNA induction in S/G2 cells. This aligns with the functions of TOP2A, such as resolving DNA catenates between sister chromatids or ending replication at repetitive sequences, that is DNA transaction dependent on the cell cycle phase (*Pommier et al., 2022*; *Lee and Berger, 2019*).

These findings suggest that the signaling of rDNA damage for PNA formation can originate from various sources. The common characteristics shared by PNA-inducing lesions are the persistence of rDNA damage, its accumulation in the nucleolar caps, and resistance to rapid repair by NHEJ.

## Topological stress and simultaneous inhibition of RNAPI are potent signals for PML nuclear interaction

As previously reported, PML interaction with the nucleolar surface is induced by various types of treatments, predominantly genotoxic (*Bernardi et al., 2004*; *Condemine et al., 2007*). Our research supports these findings and offers more detailed insights, particularly that substances causing topological stress and concurrently inhibiting RNAPI are the strongest triggers for PNAs. Furthermore, our results suggest that the signal for PNAs formation is enhanced when the treatment introduces rDNA lesions or topological alterations that inhibit RNAPI in the entire nucleolus.

By using a wide array of agents impacting various cellular processes, we found that the most effective inducers of PNAs (detectable in over 10% of nuclei) have some common characteristics: they are well-established inhibitors of topoisomerases and/or modify DNA topology, and they result in long-term inhibition of RNAPI, as evidenced by the presence of nucleolar caps 2 days after treatment.

The topological alterations that these compounds introduce into chromatin depend on different mechanisms. For example, doxorubicin impacts DNA topology in three ways: it acts as a topoisomerase poison and catalytic inhibitor, and it can also intercalate DNA. BMH-21 intercalates DNA (*Colis et al., 2014*), and this interaction with chromatin leads to the trapping of TOP2 on DNA (*Espinoza et al., 2024*). Aclarubicin, a DNA intercalator derived from doxorubicin (*Sehested and Jensen, 1996*), induces chromatin changes like BMH-21 (*Espinoza et al., 2024*). CPT and TPT are well-known poisons of TOP1. It is important to mention that both poisons are a source of R-loops in actively transcribed DNA regions, including rDNA (*Marinello et al., 2013*). R-loops are non-B DNA structures being formed by RNA-DNA hybrid. When unsolved, these structures contribute to genomic instability and can be considered as a specific type of topological aberration that can potentially trigger the PNAs. Lastly, oxaliplatin forms bulky adducts with purines, causing conformational distortions of DNA, and this topological aberration increases the expression of TOP1 (*Raymond et al., 1998*). Importantly, our hypothesis that topological stress initiates the formation of PNAs is supported by the downregulation of TOP1 and TOP2A, which also triggered PML nucleolar interaction.

It is worth noting that not all compounds cause a large number of DSBs. Oxaliplatin, BMH-21, and aclarubicin evoked a lower number of DSBs compared to doxorubicin. Moreover, the DNA damage introduced by these compounds is not easily repaired by NHEJ. As mentioned above, doxorubicin induces DNA damage, the repair of which relies on both HDR and NHEJ. An RNA interference screen used to identify genes that affect cell sensitivity to oxaliplatin revealed that defects in NHEJ, unlike deficiency in BRCA2, do not sensitize cells to oxaliplatin (*Bruno et al., 2017*). The TOP1 poisons (CPT and TPT) primarily introduce DNA SSBs; however, when DNA SSBs collide with DNA transcription or replication machineries, they can be transformed into DSBs, and again, HDR is involved in their resolution (*Arnaudeau et al., 2001*; *O'Connell et al., 2010*). Finally, BMH-21 and aclarubicin affect the structure of chromatin by intercalation, and it is not yet clear how these complex topological issues in (r)DNA are resolved (*Espinoza et al., 2024*).

The other common feature of these potent PNA-inducing treatments is long-term RNAPI inhibition associated with the formation of nucleolar caps. The interference with rRNA transcription has already been described for some of them. For example, doxorubicin restrains rRNA transcription in higher concentrations (*Burger et al., 2010*). BMH-21 has been primarily used as an inhibitor of RNAPI (*Peltonen et al., 2014*). It was found recently that aclarubicin, by its intercalation into DNA, affects more chromatin-linked processes, including rRNA transcription (*Espinoza et al., 2024*). The oxaliplatin, by an unknown mechanism, induces ribosome biogenesis stress and inhibition of rRNA transcription, which strongly contributes to its cytotoxicity (*Bruno et al., 2017*; *Schmidt et al., 2022*). CPT and TPT have not yet been described as treatments inhibiting rRNA transcription. Nevertheless, CPT affects early rRNA processing (*Burger et al., 2010*) and nucleolar morphology (*Potapova et al.,*

*2023*). Notably, we observed the formation of the nucleolar caps (and RNAPI inhibition) and the highest number of PNAs only when the rather high concentration of CPT or TPT (50 μM) was used. Thus, our data suggest that only significant TOP1 deficiency can inhibit RNAPI in the entire nucleolus. Moreover, the link between long-term TOP1 hindrance and RNAPI inhibition was observed after TOP1 knockdown, as nucleolar caps wrapped by PNAs were present 3- and 6-days post-transfection.

Thus, we think that the accumulation of topological stress and/or other chromatin (rDNA) aberrations contribute to the inhibition of pre-rRNA transcription and can efficiently trigger PNAs.

## The sole RNAPI inhibition or TOP2 poisoning are weak signal of PNAs

Notably, three additional treatments that trigger the signal for PML nucleolar interaction are also associated with RNAPI inhibition or topological stress, yet the resulting fraction of nuclei with PNAs was only around 5%. AMD and CX-5461 are primarily known as RNAPI inhibitors; etoposide is a TOP2 poison. AMD is an intercalator with a higher affinity for GC-rich DNA regions and, at the low concentration (10 nM) used in this study, inhibits RNAPI without introducing DSBs (*Sobell et al., 1971*; *Chen et al., 1996*). After 2 days of treatment, RNAPI was inhibited in all nuclei. Interestingly, the same 'nucleolar' pattern was observed after doxorubicin and other treatments that are strong inducers of PNAs. Notably, PNAs were present in only 6% of nuclei after AMD, nearly fourfold less than after doxorubicin (22%). This observation suggests that the mere formation of the nucleolar cap (inhibition of RNAPI) is not sufficient, and some additional signal/s induced by an event that happens rather rarely under AMD treatment is also involved in triggering PNAs. It was recently shown that 1 μM AMD could introduce a transition between B-DNA to Z-DNA in whole chromatin (*Espinoza et al., 2024*). Thus, we can only speculate that such changes in the conformation of rDNA can accidentally occur, albeit rarely, even when the hundred times lower concentration of AMD is used to inhibit RNAPI specifically.

The other inhibitor of RNAPI used in this study was CX-5461. Importantly, 2 days after treatment, only a fraction of RNAPI was accumulated in the nucleolar caps, indicating partial inhibition of RNAPI activity. After this treatment, PML-NDS, the PNA types linked to the reactivated nucleolus, were present in 4% of nuclei. The signals for PNAs formation after CX-5461 might reflect a partial inhibition of RNAPI in combination with TOP2 inhibition, as CX-5461 has been newly characterized as a plausible TOP2 poison (*Pan et al., 2021*).

Another question is why topological stress induced by pure TOP2 poisons (etoposide, CX-5461) provides only weak signals for the interaction of PML with the nucleolar cap. We analyzed the critical differences to better understand the respective toxicity mechanisms of etoposide and doxorubicin (triggering PNAs in 4% and 22% of cells, respectively). Etoposide is a pure TOP2 poison, which stabilizes the TOP2-DNA covalent complexes (also referred to as ternary complexes) in a broad range of concentrations and, therefore, protects the re-ligation of DNA (*Willmore et al., 1998*). Thus, the formation of DSBs is the final consequence of etoposide exposure. On the other hand, anthracycline doxorubicin stabilizes the TOP2-DNA only at low concentrations, and the number of ternary complexes is always lower compared to etoposide (*Atwal et al., 2019*). Besides the TOP2 poisoning activity, doxorubicin can inhibit TOP2-mediated decatenation (*Frank et al., 2016*; *Hasinoff et al., 2016*), intercalates DNA, and by this alters DNA torsion and induces histone eviction (*Pang et al., 2013*), and elevates oxidative stress (*Berlin and Haseltine, 1981*; *Gajewski et al., 2007*). Therefore, the DNA damage introduced by doxorubicin is more complicated than after etoposide. Based on these findings, we hypothesize that the DSBs caused by freezing TOP2 in catalytic action do not cause sufficient aberration of (r)DNA to signal for extensive PML nucleolar interaction despite the TOP2 activity being affected and the DDR being present. This observation again supports our hypothesis that for the induction of PNAs, the introduced (r)DNA damage must be more challenging to repair.

## The analogy between PNAs and APB

Our study suggests that sustained damage in the rDNA locus (NOR) can prompt the accumulation of PML on the surface of the nucleolar cap. PNAs, as a dynamic and structurally diverse PML structure, have the potential to identify and isolate damaged rDNA adjacent to the active nucleolus by forming PML-NDS. However, it remains unclear whether the sequence of structural changes from bowl- to funnel-type PNAs to PML-NDS is solely involved in sequestering damaged rDNA away from active NORs or if it is associated with a specific process related to the repair of rDNA lesions 'resistant' to NHEJ and HDR. It has been demonstrated that PML is associated with persistent DSB following

irradiation (*Vancurova et al., 2019*). The same article presented that PML-deficient (knockout) RPE-1$^{hTERT}$ cells exhibited poorer recovery from genotoxic treatment introducing DNA lesions, which HDR can only repair. This suggests an active role of PML in HDR, but the mechanism by which PML is involved in the repair of such DNA lesions has not yet been elucidated. Additionally, it has been shown that PML is crucial for the alternative lengthening of telomeres, a specific type of HR dependent on break-induced replication in cancer cells that do not express telomerase. Mechanistically, PML accumulates damaged telomeres into the APB through SUMO-SIM interaction and acts as an interface for the repair. It has been shown that the clustering of telomeres and the directing of certain proteins involved in the repair of damaged telomeres depends on PML (*Osterwald et al., 2015*; *Zhang et al., 2019*). There are several similarities between PNAs and APBs. The interaction partner of PML located on both the telomeres and rDNA must be sumoylated, as the PML-SIM domain is essential for the formation of both APBs and PNAs (*Hornofova et al., 2022*; *Chung et al., 2011*). The PML IV isoform most efficiently forms APBs and also PNAs (*Zhang et al., 2021*; *Hornofova et al., 2022*). PML clusters the damaged telomeres into APBs, and we observe that several NORs converge in one PNA structure; thus, the PML-dependent clustering of damaged NORs is plausible. On the other hand, there is one critical difference between the otherwise broadly analogous APBs and PNAs. The process of ALT operates in transformed cancer cells that do not express the telomerase, thus enabling telomere maintenance, cell proliferation, and immortalization (*Henson et al., 2005*; *Villa et al., 2008*). The PNAs, on the other hand, were primarily detected in non-transformed cells, and their formation is linked to cell cycle arrest and establishment of senescence (*Janderová-Rossmeislová et al., 2007*; *Imrichova et al., 2019*). It remains to be determined whether the formation of PNAs is positively involved in rDNA repair, resulting in a return of at least some PNA-forming cells to the cell cycle, or if they play a role in blocking the repair of DSBs on rDNA, broadly analogous to the shelterin complex on telomeres during replicative senescence (*d'Adda di Fagagna et al., 2003*). We propose that the pro-senescent role of PNAs may contribute to the maintenance of rDNA stability, thereby limiting the potential of hazardous genomic instability and, hence, the risk of cellular transformation. Analogous to checkpoint responses and oncogene-induced senescence (*Bartkova et al., 2006*; *Halazonetis et al., 2008*), the PNA-associated senescence might provide one aspect of the multifaceted cell-autonomous anticancer barrier, in this case guarding the integrity of the most vulnerable repetitive rDNA loci, possibly at the expense of accumulated cellular senescence-associated decline of functional tissues during aging.

## Conclusion

Based on the results of our present study, we propose a concept that PML nucleolar associations are initiated in response to persistent damage to nucleolar rDNA (NORs). This conclusion was reached upon examination of a variety of genotoxic treatments and other interventions. We observed that signals leading to the generation of PNAs (i) rely on the activities of ATM and ATR kinases, and (ii) are induced by direct rDNA damage and modulation of the DNA repair pathways, including those dependent on homology-directed repair. While the precise molecular role of PNAs in addressing these rDNA impairments remains to be fully elucidated, we propose that PNAs identify damaged rDNA and, by facilitating the development of PML-NDS adjacent to the active nucleolus, they can segregate aberrant rDNA from undamaged NORs. It is important to highlight that PNAs are seldom seen in tumor cells. Whether their absence contributes to the characteristic rDNA instability in tumor cells remains an unresolved question. The identification of chemotherapeutic drugs capable of introducing persistent damage into the rDNA locus (NORs) and causing long-term RNAPI inhibition, thereby sensitizing cells to cell death or cellular senescence, opens new avenues for research in exploring the synergistic effects of new combinations of topoisomerase and RNA polymerase I inhibitors, especially in the context of cancer therapy.

## Materials and methods
### Chemicals and antibodies

4',6-Diamidino-2-phenylindole (DAPI; D9542), aphidicolin (A0781), actinomycin D (50-76-0), 5-bromo-2´-deoxyuridine (B5002), BMH-21 (SML1183), camptothecin (C9911), CX-5461 (1138549-36-6), DIG-Nick Translation Mix (11745816910), dimethyl sulfoxide (D2650), doxorubicin hydrochloride

(D1515), doxycycline hyclate (D9891), etoposide (E1383), 5-fluorouracil (F6627), 5-fluorouridine (F5130), G418 disulfate salt (G5013), hydroxyurea (H8627), MG-132 (C2211), oxaliplatin (O9512), RAD51 Inhibitor B02 (SML0364), roscovitine (R7772) and topotecan hydrochloride (T2705) were all purchased from Sigma-Aldrich/Merck (Darmstadt, Germany). Aclarubicin (A2601) was obtained from APExBIO (Houston, TX, USA), Annexin V-FITC (EXB0024) from EXBIO Praha, a.s. (Vestec, Czech Republic), pyridostatin (4763) from TOCRIS (Bristol, United Kingdom), and the interferon gamma (IFNγ) recombinant protein (300-02) from Peprotech (Rocky Hill, NJ, USA), TOTO-3 (T-3604). KU-60019 (S1570), KU-55933 (S1092), and VE-822 (S7102) were purchased from Selleckchem (Cologne, Germany). Lipofectamine RNAiMAX Reagent (13778075) and ProLong Gold Antifade Mountant (P36934) were obtained from Thermo Fisher Scientific (Waltham, MA, USA), NU 7447 (3712) was from BioTechne/NovusBiological (Minneapolis, USA), Shield-1 (AOB1848) from AOBIOUS (Gloucester, MA, USA). Nick translation DNA labeling system 2.0 (ENZ-GEN111-0050), and Red 650 [Cyanine-5E] dUTP (ENZ-42522) from Enzo Biochem (New York, NY, USA), Hybrisol VII (ICNA11RIST139010) from VWR International (Singapore).

Specification of primary and secondary antibodies used throughout the study is listed in **Supplementary file 2**.

## Cell culture

Immortalized human retinal pigment epithelial cells (RPE-1$^{hTERT}$, ATCC) were cultured in Dulbecco's modified Eagle's medium (Gibco/Thermo Fisher Scientific, Waltham, MA, USA) containing 4.5 g/L glucose and supplemented with 10% fetal bovine serum (Gibco/Thermo Fisher Scientific, Waltham, MA, USA) and antibiotics (100 U/mL penicillin and 100 μg/mL streptomycin sulfate, Sigma, St. Louis, MO, USA). The cells were cultured in normal atmospheric air containing 5% $CO_2$ in a standard humidified incubator at 37 °C on a tissue culture dish (TPP Techno Plastic Products AG, Trasadingen, Switzerland). For treatments, see **Supplementary file 1**. For IR exposure, the cells were irradiated with orthovoltage X-ray instrument T-200 (Wolf-Medizintechnik) using a dose of 10 Gy. The cells with inducible expression of endonuclease I-PpoI were derived from RPE-1$^{hTERT}$ (the detailed protocol of their generation follows). The culture conditions were the same as for RPE-1$^{hTERT}$, except the certified TET-free 10% fetal bovine serum (Gibco/Thermo Fisher Scientific, Waltham, MA, USA) was used to repress the expression of I-PpoI. The seeding concentration for all experiments was 20,000 cells/cm². The 0.5 μM Shield-1 and 1 μM doxycycline were used to induce I-PpoI. The 10 μM B02 and 1 μM NU-7441 were added simultaneously with doxorubicin or Shield and doxycycline to inhibit HR or NHEJ, respectively. The 6 μM KU-55933, 1 μM KU-60019, and 0.2 μM VE-822 were added 1 hr before I-PpoI activation or doxorubicin induction to inhibit ATM, ATM, or ATR, respectively. The employed concentration of ATM and ATR inhibitors are commonly used in the DNA damage field (**Zeng et al., 2023**; **Negi and Brown, 2015**; **Golding et al., 2009**; **Ortega-Atienza et al., 2016**; **Zhan et al., 2010**; **Hickson et al., 2004**).

## Generation of cell lines with inducible expression of I-PpoI

All plasmids and primers used in this study are listed in **Supplementary file 3** and **Supplementary file 4**, respectively. To generate a cell line with the regulatable expression of endonuclease I-Ppo-I, the lentiviral plasmid pLVX-TetOne-neo bearing a cassette with endonuclease I-PpoI fused to destabilization domain FKBP and HA-tag (pLVX-TETOne-neo-FKBP-PPO-HA) was prepared as follows. Phusion High-Fidelity DNA Polymerase (M0530L) was used for all DNA amplification steps. DNA fragment with FKBP-I-Ppo-I-HA was amplified from plasmid pCDNA4TO-FKBP-PPO-HA (**Warmerdam and Wolthuis, 2019**) using primers GA-PpoI-LVXpur-fr-F and GA-PpoI-LVXpur-fr-R and then, using Gibson assembly, was inserted into pLVX-TETOne-neo plasmid cleaved by BamHI, EcoRI and dephosphorylated by Shrimp Alkaline Phosphatase (EF0511, Fermentas). The correct fusion was verified by sequencing. The plasmid pLVX-TET-ONE-neo was prepared by exchange of region coding for puromycin N-acetyl-transferase (*puromycin resistance*) with DNA fragment coding for neomycin phosphotransferase (*npt; neomycin resistance*). First, the core of pLVX-TETOne-Puro was linearized by PCR using primers GA-V-LVXpuro-F and GA-V-LVXpuro-R, and the DNA fragment with *npt* gene was obtained by PCR amplification of plasmid pCDH-CMV-MCS-EF1-Neo using primers GA-F-neo-R and GA-F-neo-F. Finally, both fragments were assembled by Gibson assembly, and the correct fusion was verified by sequencing. The stable RPE-1$^{hTERT}$ cells with doxycycline-inducible expression of FKBP-PpoI-HA were generated

by lentiviral infection and subsequent selection with G418 (1.12 mg/mL). Afterward, the resistant cells were seeded into the 96-well plate to isolate single-cell clones. The generated cell lines were inspected for mycoplasma contamination and then for the expression of I-PpoI.

## Indirect immunofluorescence, high-content, and confocal microscopy

Cells grown on glass coverslips were fixed with 4% formaldehyde in PBS for 15 min, permeabilized in 0.2% Triton X-100 in PBS for 10 min, blocked in 10% FBS in PBS for 30 min, and incubated with primary antibodies for 1 hr, all in RT. After that, cells were washed thrice in PBS for 5 min, and secondary antibodies were applied in RT for 1 hr. For some experiments, 1 µM TOTO-3 was applied together with secondary antibodies. Subsequently, cells were counterstained with 1 µg/mL DAPI for 2 min, washed thrice with PBS for 5 min, and let dry. After that, coverslips were mounted in Prolong Gold Antifade mountant. To detect RAD51 and phosphorylated form of RPA32 (pS33), a pre-extraction step was added: before formaldehyde fixation cells were incubated on ice in pre-extraction buffer (25 mM HEPES, pH 7.7, 50 mM sodium chloride, 1 mM EDTA, 3 mM magnesium chloride, 300 mM sucrose, 0.5% Triton X-100) for 5 min and after fixation cells were washed with pre-extraction buffer without Triton X-100. The rest of the protocol remains the same. The wide-field images were acquired on the Leica DM6000 fluorescent microscope using the HCX PL APO 63×/1.40 OIL PH3 CS and HCX PL APO 40×/0.75 DRY PH2 objectives and monochrome CCD camera Leica DFC 350 FX (Leica Microsystems GmbH, Wetzlar, Germany); or on the Nikon Eclipse Ti2 Inverted Microscope with CFI PL APO 60×/1.40 OIL PH3 objective and DS-Qi2 high-sensitivity monochrome camera Andor Zyla VSC-07008. High-content image acquisition (ScanR) was made on the Olympus IX81 microscope (Olympus Corporation, Tokyo, Japan) equipped with ScanR module using the UPLFN 40×/1.3 OIL objective and sCMOS camera Hamamatsu ORCA-Flash4.0 V2 (Hamamatsu Photonics, Shizuoka, Japan). The data were analyzed using ScanR Analysis software (Olympus Corporation, Tokyo, Japan). High-resolution images were captured by a confocal Leica STELLARIS 8 microscope equipped with HC PL APO 63×/1.40 OIL objective using type F immersion oil (Leica, 11513859), DAPI was excited by 405 nm laser, whereas Alexa Fluor 488, Alexa Fluor 568 and TOTO-3 were excited by a white light laser tuned to 499 nm, 579 and 642 nm, respectively, and the signal was bidirectionally sequentially scanned with the use of hybrid detectors. Images were acquired as a Z-stacks with sufficient number of 0.2 µm step to cover the whole nuclei. The images were acquired and processed with Leica LIGHTNING deconvolution set to maximal resolution, but with the pinhole open to 1 AU and zoom set to 8, large nuclei were rotated diagonally. The images obtained by immune-FISH and used for the analysis of the link between rDNA/DJ DNA locus and PML and DNA damage were captured by confocal laser scanning inverted microscope DMi8 with confocal head Leica TCS SP8 and equipped by HC PL APO 63×/1.40 OIL CS2; FWD 0.14; CG 0.17 | BF, POL, DIC objective using type F immersion oil (Leica, 11513859). The Alexa Fluor 408 was excited by UV laser (405 nm), whereas Alexa Fluor 488, rhodamine, and enhanced Cyanine 5-dUTP were excited by a white light laser tuned to 499 nm, 552 nm and 638 nm, respectively. The signal was bidirectionally sequentially scanned, and images were acquired as a Z-stacks with sufficient number of 0.2 µm step to cover whole nuclei. The obtained images were deconvoluted using Huygens Professional 20.10 software. All images were processed using Fiji/ImageJ2 software (*Rueden et al., 2017*; *Schindelin et al., 2012*). The Fiji plugin Mosaic/Squassh (Squassh – segmentation and quantification of subcellular shapes) was used for co-localization analysis. The software was used according to the protocol and guidelines recommended by the authors (*Rizk et al., 2014*). A co-localization mask was generated to prepare 3D co-localization models as an overlap of individual masks generated by Squassh. Co-localization mask and the masks of individual channels were then upscaled by factor 10, and their surfaces were visualized in Imaris software (Oxford Instruments).

## rDNA immuno-FISH

For immuno-FISH, the procedure described previously was used (*Mais et al., 2005*). Briefly, RPE-1$^{hTERT}$ cells were grown on Superfrost Plus slides (R886761, P-lab), washed with PBS, fixed with 4% formaldehyde at RT for 10 min, and washed 3×10 min with PBS. After that, the cells were permeabilized with 0.5% Triton X-100/PBS for 10 min at RT, washed 3×10 min with PBS, and incubated in 20% glycerol/PBS for 2 hr . The slides were then snap-frozen in liquid nitrogen and stored at –80 °C. To produce fluorescent FISH probes, plasmid (for rDNA probe, *Supplementary file 3*) or BAC (for DJ probe, *Supplementary file 3*) DNA was labeled by nick-translation according to the manufacturer's instructions.

The rDNA probe binds a 12 kb segment of rDNA intergenic spacer (immediately upstream of the promoter), while the DJ probe covers the interval between 76.5 kb and 259.4 kb distal to rDNA. For labeling, the slides were rinsed with PBS for 3×10 min, washed briefly with 0.1 M HCl, incubated in 0.1 N HCl for 5 min, washed in 2×SSC (30 mM sodium citrate, 300 mM sodium chloride, pH 7) for 5 min, and let dry. The slides were then incubated with equilibration buffer (50% formamide/2×SSC) for 15 min at 37 °C. After 15 min, a DNA probe in Hybrisol VII was applied to clean coverslips, the equilibration buffer was shaken off the slides with cells, and the slides were lowered on the coverslips with the probe. The slides were then sealed, denatured for 12 min at 73 °C on a heating block, and incubated at 37 °C for 16–48 hr. After that, the coverslips were removed, and the slides were washed 3×5 min with 50% formamide/2×SSC at 42 °C; and 3×5 min with 0.1×SSC preheated to 60 °C. Then, the slides were washed 3×5 min with PBS and IF staining, and microscope imaging was performed as described above.

## 5-FU incorporation assay

For the 5-FU incorporation assay, cells were incubated with 1 mM 5-FU for 30 min at indicated time points after doxorubicin treatment and removal. After that, cells were fixed with 4% formaldehyde at RT for 15 min, and the 5-FU incorporation was visualized using anti-BrdU antibody cross-reacting with 5-FU. The standard protocol for immunofluorescence described above was used.

## RNA interference

For TDP2, cells were plated on six-well plates 1 day before siRNA transfection, and 30 pmol of siRNA per well was used. For TOP1, TOP2A, TOP2B, RAD51, and LIG4, the reverse transfection was used. The cells were transfected with 20 pmol of si/esiRNAs during the seeding on six-well plates. The siRNA targeting TDP2 (5' CCUAUGUUGACCUAACCAAtt 3'; PDSIRNA2D; Sigma-Aldrich/Merck), siRNAs targeting TOP2B (5' CGAUUAAGUUAUUACGGUUtt 3', s106; 5' GAGUGUACACUGAUAUUAAtt 3', s108; Ambion/ThermoFisher Scientific), and MISSION esiRNA (Sigma-Aldrich/Merck; Darmstadt, Germany) targeting TOP1 (EHU101551), TOP2A (EHU073241), RAD51 (EHU045521), and LIG4 (EHU062841) were used. As control the non-targeting siRNA (Silencer Select Negative Control No. 1, 4390843; Ambion/ThermoFisher Scientific) and ON-TARGET plus Non-targeting Control Pool (D-001810-10-05, Dharmacon) were applied. The transfections were performed using Lipofectamine RNAiMAX, according to the manufacturer's instructions.

## SDS-PAGE and immunoblotting

Cells were harvested into SDS sample lysis buffer (62.5 mM Tris-HCl, pH 6.8, 2% SDS, 10% glycerol), boiled at 95 °C for 5 min, sonicated and centrifuged at 18,000 × g for 10 min. The BCA method estimated protein concentration (Pierce Biotechnology Inc, Rockford, USA). Equal amounts of total protein were mixed with DTT and bromphenol blue to a final concentration of 100 mM and 0.01%, respectively, and separated by SDS-PAGE (8%, 12%, or 4–12% gradient polyacrylamide gels were used). The proteins were electrotransferred to a nitrocellulose membrane using wet or semi-dry transfer. Immunostaining followed by ECL detection was performed. The intensity of the bands was measured in Fiji/ImageJ Gel Analyzer plugin, and the protein levels were calculated as the band intensities of the proteins of interest, related to the band intensities of loading control, while the relative intensity of untreated cells (or cells treated with a dissolvent – acetic acid or DMSO) was set as one.

## Quantitative real time PCR (qRT-PCR)

Total RNA samples were isolated using RNeasy Mini Kit (74134, QIAGEN, MD, USA) according to the manufacturer's protocol. First strand cDNA was synthesized from 400 ng of total RNA with random hexamer primers using TaqMan Reverse Transcription Reagents (N8080234, Applied Biosystems/Thermo Fisher Scientific, Waltham, MA, USA). qRT-PCR was performed in LightCycler 480 System (Roche, Basel, Switzerland), using LightCycler 480 SYBR Green I Master (04887352001, Roche, Basel, Switzerland). The relative quantity of cDNA was estimated by ΔΔCt *Livak and Schmittgen, 2001*; while the GAPDH was used as a reference (*Vandesompele et al., 2002*).

## Detection of apoptosis

RPE-1$^{hTERT}$ clones 1A11 and 1H4 were seeded on a six-well plate (20,000 cells/cm$^2$). The following day, expression of I-PpoI endonuclease was induced, or adequate volume of vehicle control (DMSO) was added for 24 hr and harvested (floating, wash, and adherent fractions were pooled) and centrifuged at 200 × $g$ for 3 min. The cell pellet was washed in PBS and centrifuged at 200 × $g$ for 3 min and resuspended in 200 μL of Annexin V binding buffer (Apronex, the Czech Republic) containing 1 μL annexin V-FITC (Exbio, the Czech Republic) and incubated at RT for 30 min . After incubation, 1 ml of Annexin V binding buffer was added, and the cell suspension was centrifuged at 200 × $g$ for 3 min. The cell pellet was resuspended in 200 μL of Annexin V binding buffer. Cells were counted by flow cytometer BD LSRII (BD Biosciences, San Jose, USA). The acquired data were analyzed using FlowJo software (Tree Star, Ashland, USA).

## Colony formation assay (CFA)

RPE-1 $^{hTERT}$ clones 1A11 and 1H4 were seeded on 24-well plates (10,000 cells/cm$^2$) and allowed to adhere overnight. The following day, the expression of I-PpoI endonuclease was induced, or adequate volume of vehicle control (DMSO) was added. After 24 hours of treatment, cells were collected via trypsinization, counted, and reseeded in triplicates into a 60 mm$^2$ cell culture dish in DMEM. The seeding density for control cells was 200 cells per dish and 4000 cells per dish for treated cells. Cells were maintained for 7 days to allow colony formation. The experiment was done in triplicate. Cells were fixed with a staining solution (0.05% w/v crystal violet, 1% formaldehyde, 1% methanol) for 20 min, washed with water, air dried, and scanned by Epson Perfection V700 Photo scanner (Epson, Amsterdam, Netherlands). Colonies were analyzed using Fiji/ImageJ-based macro. Images of individual colonies were taken on ZEISS AxioZoom V.16 microscope using the objective PlanNeoFluar Z 1.0 x/0.25 and ZEISS Axiocam 305 – color CMOS camera.

## Senescence-associated β-galactosidase activity

The development of cellular senescence was followed by the determination of senescence-associated β-galactosidase activity (***Dimri et al., 1995***). RPE-1$^{hTERT}$-I-PpoI cells (clone 1H4) were seeded on 24-well plates (10,000 cells/cm$^2$) and allowed to adhere overnight. The following day, the expression of I-PpoI endonuclease was induced. After 24 hr of treatment, cells were collected via trypsinization, counted, and reseeded on coverslips into a 60 mm$^2$ cell culture dish. The seeding density was 10,000 cells per dish and fixed 8 and 12 days later with 0.5% glutaraldehyde at RT for 15 min. Cells were washed twice with 1 mM MgCl$_2$/PBS and incubated with X-Gal staining solution at 37 °C for 16 hr. The staining was terminated by three consecutive washes with PBS. Finally, the cells were mounted with Mowiol (81381, Sigma-Aldrich/Merck) containing DAPI and imaged on the Leica DM6000 fluorescent microscope using the HC PLAN APO 20×/0.70 DRY PH2 objective and color CCD camera Leica DFC490 (Leica Microsystems GmbH, Wetzlar, Germany).

## Estimation of G1, S, and G2 cells after doxorubicin treatment

Twenty-four h after seeding of RPE-1$^{hTERT}$ cells, doxorubicin (0.75 μM) and EdU (10 μM) were added. Forty-eight h after treatment, the cells were harvested, and PML was detected by indirect IF and EdU using a kit (Click-iT EdU Alexa Fluor 647 Imaging Kit; C10340; Invitrogen) according to the manufacturer's instructions. High-content image acquisition and analysis were used to estimate G1, S, and G2 cells according to the total DAPI fluorescence and EdU positivity.

## Acknowledgements

This study was supported by the Grant Agency of the Czech Republic (Project 19–21325 S and 23–07273 S), and the Institutional Grant (Project RVO 68378050). ZH and PV were supported in part by the project National Institute for Cancer Research (Programme EXCELES, ID Project No. LX22NPO5102) – Funded by the European Union – Next Generation EU. JB was supported in part by the Danish Cancer Society grant (R322-A17482), The Danish Council for Independent Research grant (1026-00241B), and the Novo Nordisk Foundation grant (NNF20OC0060590). We thank Mikael K Lindstrom for helpful discussions. We acknowledge the Light Microscopy Core Facility, IMG, Prague, Czech Republic, supported by grants "National Infrastructure for Biological and Medical Imaging"

(MEYS – LM2023050), "Modernization of the national infrastructure for biological and medical imaging Czech-BioImaging" (MEYS – CZ.02.1.01/0.0/0.0/18_046/0016045) and formal National Program of Sustainability NPUI LO1220 and LO1419 (RVO: 68378050-KAV-NPUI), for their support with the confocal, widefield imaging, and image analysis presented herein. We thank Marketa Vancurova, Martin Capek, Jan Valecka, Ivan Novotny, Helena Chmelova, and Jiri Cerny for their excellent technical support.

## Additional information

### Funding

| Funder | Grant reference number | Author |
|---|---|---|
| Grant Agency of the Czech Republic | 19-21325S | Jiri Bartek |
| Grant Agency of the Czech Republic | 23- 07273S | Jiri Bartek |
| Institutional Grant Project RVO | 68378050 | Zdenek Hodny |
| National Institute for Cancer Research; Programme EXCELES, | LX22NPO5102 | Zdenek Hodny |
| Danish Cancer Society | R322-A17482 | Jiri Bartek |
| Danish Council for Independent Research | 1026-00241B | Jiri Bartek |
| Novo Nordisk Foundation | NNF20OC0060590 | Jiri Bartek |

The funders had no role in study design, data collection and interpretation, or the decision to submit the work for publication.

### Author contributions

Alexandra Urbancokova, Terezie Hornofova, Conceptualization, Investigation, Visualization, Methodology, Writing - original draft; Josef Novak, Investigation, Visualization; Sarka Andrs Salajkova, Methodology; Sona Stemberkova Hubackova, Conceptualization, Investigation; Alena Uvizl, Tereza Buchtova, Investigation; Martin Mistrik, Funding acquisition, Methodology, Writing – review and editing; Brian McStay, Methodology, Writing – review and editing; Zdenek Hodny, Conceptualization, Resources, Supervision, Funding acquisition, Project administration, Writing – review and editing; Jiri Bartek, Conceptualization, Supervision, Funding acquisition, Writing – review and editing; Pavla Vasicova, Conceptualization, Supervision, Validation, Investigation, Visualization, Methodology, Writing - original draft, Project administration, Writing – review and editing

### Author ORCIDs

Brian McStay ⓘ https://orcid.org/0000-0002-5664-7781
Pavla Vasicova ⓘ https://orcid.org/0000-0002-9733-9929

Reviewer #1 (Public review): https://doi.org/10.7554/eLife.91304.3.sa1
Author response https://doi.org/10.7554/eLife.91304.3.sa2

## Additional files

### Supplementary files

• Supplementary file 1. The compounds/treatments tested for the ability to induce PML-nucleolar association. The applied concentration and toxicity specification are shown.

• Supplementary file 2. The antibodies, including description, source, and dilution, are shown.

• Supplementary file 3. The names of plasmids and BACs, as well as descriptions and sources, are

shown.

- Supplementary file 4. The sequence of primers, as well as their application, are shown.
- MDAR checklist

### Data availability

All source numerical data files used for graphs and histograms together with all original membranes for western blots have been provided.

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
