## [Editor Report · eLife assessment]

This **valuable** study asks how Promyelocytic leukemia protein (PML) becomes associated with the nucleoli of cells (PML Nucleolar Associations, PNAs) upon various genotoxic stimuli. Using immunostaining analysis with induced DNA double-strand breaks (DSBs) in rDNA repeats, the authors provide **solid** evidence that PNAs are triggered mostly by the inhibition of topoisomerase and RNA polymerase I, which is augmented by homologous recombination but not by the non-homologous end joining double-strand break repair pathway. The findings have potential implications for a better understanding of how DNA damage in ribosomal DNA is repaired for genome stability. This paper is of interest to researchers in the fields of nuclear structure and DNA repair.

---

## [Referee Report · Reviewer #1 (Public review)]

Summary:

This paper described the dynamics of the nuclear substructure called PML Nucleolar Association (PNA) in response to DNA damage on ribosomal DNA (rDNA) repeats. The authors showed that the PNA with rDNA repeats is induced by the inhibition of topoisomerases and RNA polymerase I and that the PNA formation is modulated by RAD51, thus homologous recombination. Artificially induced DNA double-strand breaks (DSBs) in rDNA repeats stimulate the formation of PNA with DSB markers. This DSB-triggered PNA formation is regulated by DSB repair pathways.

Strengths:

This paper illustrates a unique DNA damage-induced sub-nuclear structure containing the PML body, which is specifically associated with the nucleolus. Moreover, the dynamics of this PML Nucleolar Association (PNA) require topoisomerases and RNA polymerase I and are modulated by RAD51-mediated homologous recombination and non-homologous end-joining. This study provides a unique regulation of DSB repair at rDNA repeats associated with the unique-membrane-less subnuclear structure.

Weaknesses:

Although the PNA formation on rDNA repeat is nicely shown by cytological analysis, the biological significance of PNA in DSB repair is not fully addressed.

---

## [Author Response]

The following is the authors’ response to the original reviews.

**Reviewer #1 (Public Review):**
Summary:This paper described the dynamics of the nuclear substructure called PML Nucleolar Association(PNA) in response to DNA damage on ribosomal DNA (rDNA) repeats. The authors showed that the PNA with rDNA repeats is induced by the inhibition of topoisomerases and RNA polymerase I and that the PNA formation is modulated by RAD51, thus homologous recombination. Artificially induced DNA double-strand breaks (DSBs) in rDNA repeats stimulate the formation of PNA with DSB markers. This DSB-triggered PNA formation is regulated by DSB repair pathways.Strengths:This paper illustrates a unique DNA damage-induced sub-nuclear structure containing the PML body, which is specifically associated with the nucleolus. Moreover, the dynamics of this PML Nucleolar Association (PNA) require topoisomerases and RNA polymerase I and are modulated by RAD51mediated homologous recombination and non-homologous end-joining. This study provides a unique regulation of DSB repair at rDNA repeats associated with the unique-membrane-less subnuclear structure.Weaknesses:Although the PNA formation on rDNA repeat is nicely shown by cytological analysis, the biological significance of PNA in DSB repair is not fully addressed.

We appreciate the succinct summary, and thank you for pointing out this insightful comment. Our data show that the dynamic interaction of PML with nucleolar caps can recognize and sequester damaged rDNA from the reactivated nucleolus. We propose that through this process, the actively transcribed intact rDNA is protected from possible detrimental interaction with the defective, PNAs-sequestered rDNA, most likely to avoid the harmful intra- and inter-chromosomal recombination events that would otherwise likely occur during recombinational repair of the damaged rDNA, as the rDNA repeats present on five chromosomes are highly repetitive. Thus, this novel sorting mechanism might help sustain the integrity of repetitive rDNA loci.

Our data also indicate that the emergence of PNAs coincided with cell cycle arrest and preceded the establishment of cellular senescence. The senescent response to rDNA damage can primarily protect the genome from the instability of rDNA loci in a manner broadly analogous to that described for protecting the telomeric loci. This notion is supported by the lack of PNA formation in most cancer cells. In the broader context of the biological significance of cellular senescence at the organismal level, such robust response to hazardous rDNA damage in the individual affected cells may limit/prevent the sporadic occurrence of early cancerous lesions, at the expense of potential tissue adverse effects accumulating over time and thereby eventually contributing to organismal aging.

**Reviewer #2 (Public Review):**
In this manuscript, the authors aim to study the PML-nucleoli association (PNAs) by different genotoxic stress and to determine the underlying molecular mechanisms.First, from a diverse set of genotoxic stress conditions (topoisomerases, RNA Pol I, rRNA processing, and DNA replication stress), the authors have found that the inhibition of topoisomerases and RNA Polymerase I has the highest PNA formation associated with p53 stabilization, gamma-H2AX, and PAF49 segregation. It was further demonstrated that Rad51-mediated HR pathway but not NHEJ pathway is associated with the PNA formation. Immuno-FISH assays show that doxorubicin induces DSBs (53BP1 foci) in rDNA and PNA interactions with rDNA/DJ regions. Furthermore, endonuclease IPpol induced DSB at a defined location in rDNA and led to PNAs.Most claims by the authors are supported by the data provided. However, below weaknesses/concerns may need to be addressed to improve the quality of the study.(1) Top2B toxin doxorubicin had the highest degree of elevating PNAs; however, Top2B-knockdown had almost no noticeable effects on PNAs. How to reconcile the different phenotypes targeting Top2B?

We thank the reviewer for this comment and believe we can reconcile the results from doxorubicin treatments and the downregulation of TOP2A and B.

The different phenotypes can reflect the fact that doxorubicin targets both human TOP2 isoforms: TOP2A and TOP2B. Hence this treatment can limit any potential redundant roles of the individual topoisomerase subtypes, which, on the other hand, can be manifested under conditions when only one specific member is depleted genetically. On the other hand, it is also crucial to note that these isoforms are not fully functionally redundant. Each isoform reveals a characteristic expression pattern and distinct yet overlapping function (e.g. Nitiss J 2009, doi.org/10.1038/nrc2608, or Uusküla-Reimand 10.1126/sciadv.add4920). Thus, doxorubicin treatment or TOP2A KD can, contrary to TOP2B KD, trigger the formation of PNAs.

Additionally, besides topoisomerase inhibition and poisoning, doxorubicin intercalates DNA and elevates oxidative stress. Therefore, the observed effect of doxorubicin may also reflect, to some extent, its broader damaging impact on (r)DNA. On the other hand, the downregulation of individual topoisomerase isoforms shows how the restriction of their respective specific function/s may evoke (r)DNA damage.

(2) To test the role of Rad51 and DNA-PKcs in the PNA formation, Rad51 inhibitor B02 and DNA-PKcs inhibitor NU-7441 were chosen to use in the study. To further exclude the possible off-target of B02 and NU-7441, siRNA-mediated knockdown of Rad51 and DNA-PKcs would be an appropriate complementary approach to the pharmaceutical inhibitor approach.

We followed this stimulating suggestion, and in the revised manuscript, we used pools of siRNAs (esiRNA) to target the mRNA of RAD51 or ligase IV (LIG4) - to mimic the Rad51 chemical inhibitor B02 and the NHEJ (DNA PK) inhibitor NU-7441, respectively. The relevant new data are presented in Figure 5F-I, 6E, and F, Supplementary Figure 5D, E, F – H, and Supplementary Figure 6C-E. Notably, the results of rDNA damage triggered PNAs formation obtained using the chemical inhibition of the repair pathways and the genetic approach (knockdown), were largely consistent, thereby supporting our original conclusions. There was one interesting partial difference when the B02 RAD51 inhibitor was compared with RAD51 knockdown, which we also comment on below, and suggest a plausible explanation reflecting the fact (known for other DDR proteins such as PARP1, etc.) that the functional inhibition of an expressed protein (here RAD51, by B02) may not necessarily phenotypically recapitulate the absence of such protein (here RAD51 knockdown). Overall, we agree that this was a very important set of control experiments, in addition extended to cell cycle phase analysis.

First, the LIG4 knockdown impacted the I-PpoI-induced PNAs formation in a way that followed the same trend as the effects caused by the NHEJ pathway inhibitor NU-7441, namely increased frequency of PNAs formation when NHEJ was impaired (Figure 5E a 5I). This was expected based on what we know about the PNA formation, as the NHEJ pathway is active throughout the cell cycle, and when such repair mode is not available in the nucleolus, then more rDNA breaks remain unrepaired and must be transported to the nucleolar caps to be processed by the HR pathway, thereby also leading to more PNAs structures formed under such conditions. In terms of cell cycle phases, the observed increase of I-PpoI-induced PNAs in cells with depleted LIG4 was more pronounced in S/G2 cells, when the PNAspromoting, cap-associated HR pathway is more active. Furthermore, the enhanced occurrence of IPpoI-induced PNAs in cells depleted of LIG4 was counter-acted (partly ‘rescued/prevented’) by the concomitant treatment with the RAD51 inhibitor B02 (Figure 5E and I) compare cells with esiLIG4 alone versus esiLIG4 + B02, overall consistent with the notion that cap-associated HR pathway facilitates PNAs formation.

Second, in the analogous scenario of comparing the impact of the RAD51 chemical inhibitor (B02) with the siRNA-mediated knockdown of RAD51, the observed trends in terms of the resulting frequencies of I-PpoI-induced PNAs, were also largely consistent, in that both strategies of interfering with RAD51 resulted in fewer PNAs formed than than in cells deficient in NHEJ. On the other hand, we must stress that after RAD51 knockdown, we did not observe a decline of PNAs compared to control cells, which was detected after B02 treatment (Figure 5E and I). However, when specifically considering the cell cycle position of the individual cells, these new analyses revealed again important similarities between the knockdown and chemical inhibition of RAD51 (Figure 6E, Supplementary Figure 6E).

Before discussing the partial, cell-cycle-related difference between the impact of RAD51 chemical inhibition vs. knockdown, it is important to consider the PNAs patterns seen in cells with activated IPpoI and proficient in both, NHEJ and HR. Thus, the overall frequency of I-PpoI-induced PNAs formation was higher in G1 than in S/G2 cells. Considering that persistent rDNA DSBs trigger the formation of PNAs, this result may reflect the very limited HDR during G1 phase, in contrast to more efficient repair of I-PpoI-induced rDNA DSBs in S/G2, the cell cycle phase in which the activity of both NHEJ and HDR operate in parallel, the latter pathway offering a safer, error-free mechanism of DSB repair.

Notably, when comparing the PNAs formation frequency in cells treated with either chemical inhibition of RAD51 (with B02) or upon knockdown of RAD51, we strikingly observed that the decrease of I-PpoIinduced PNAs formation upon RAD51 knockdown was apparent only for cells in G1 (Figure 6E, and Supplementary Figure 6E). We believe that the distinct impact of RAD51 knockdown compared with that of RAD51 inhibitor (mainly seen when S/G2 cells were analyzed separately) might reflect one or a combination of several factors, including e.g. the following:

i) The knock-down-induced absence of RAD51 protein may allow access to the persistent DSB lesions by other alternative repair proteins (such as the RAD52-mediated repair reported in diverse pathophysiological circumstances including in cells undergoing senescence, a scenario very relevant for our present study). Such altered stoichiometry of proteins interacting with the persistent rDNA DSBs may contribute to the pattern of PNAs formation that is then distinct from the pattern seen in the presence of Rad51;

ii) Another difference that we observe is the somewhat enhanced frequency of ‘spontaneous’ (i.e. even without activating the I-PpoI) PNAs formation when RAD51 is depleted, a phenomenon not seen when control non-targeting siRNA is transfected or when RAD51 is acutely inhibited by B02 (Figure 5H). Such spontaneous baseline PNA formation likely reflects the enhanced persistence of unrepaired endogenously occurring DNA lesions that are already suboptimally processed during the period following the esiRNA transfection, i.e. under stepwise depletion of the RAD51 protein which is normally required to deal with such omnipresent endogenous lesions occurring during e.g. DNA replication or some oxidative/metabolic processes;

iii) The knockdown approach, while clearly robustly depleting RAD51 protein levels (see Supplementary Figure 5D) may nevertheless leave a small residual fraction of the RAD51 protein present in the cells, thereby possibly inhibiting the HDR pathway to a slightly lesser degree than the B02 inhibitor;

iv) Additionally, it should be noted that the baseline levels of I-PpoI-induced PNAs formation are somewhat higher in the transfection experiments (i.e. when using any siRNA, even the nontargeting control siRNA), compared with the less ‘invasive’ experiments of simply adding a drug/solvent to the cell culture medium. This phenomenon adds to the commonly seen (over decades, by us and many others.) above-baseline transient stress in cells exposed to transfections, often causing even moderate transient DNA damage response. Specifically, in control experiments, the level of I-PpoI-induced PNAs was around 15% in cells transfected with non-targeting siRNA, while the comparable experiment of only I-PpoI induction under non-transfection conditions was around 10%. In other words, the somewhat enhanced baseline counts of I-PpoI-induced PNAs seen in the knock-down experiments compared with chemical inhibitor experiments reflect partly the shift of the total readout counts due to the different baseline counts. This, however, does not alter the observed overall trends that are consistent in both types of experiments.

While the potential interpretation(s) of the above results are presented in the Discussion section of the revised manuscript, the full mechanistic elucidation of the impact of various experimental manipulations on the PNA formation during the cell cycle would require a dedicated follow-up study.

(3) Several previous studies have shown the activation of the nucleolar ATM-mediated DNA damage response pathway by I-Ppol-induced DSBs in rDNA. What is the role of nucleolar ATM in the regulation of PNAs?

We agree this is an important issue the solution of which (explained below) strengthens the mechanistic insights provided in our revised manuscript, and we are grateful to the reviewer for raising this question. To address this important point and even extend the scope from ATM also to ATR, we employed two small-molecule inhibitors of ATM (KU-60019 and KU55933) and also one inhibitor of ATR (VE-822), at concentrations commonly used in analogous studies in the DNA damage response field, to examine their impact on rDNA damage/PNA formation induced by I-PpoI. The new data are shown in Figures 5A and B. We found that the inhibition of either of the two kinases alone, robustly reduced the number of nuclei with PNAs, indicating that the activity of each of these two DNA damage signaling kinases is required for the formation of I-PpoI-induced PNAs in response to rDNA damage. Future experiments should elucidate precisely which of the very wide range of ATM/ATR substrates and/or specific protein domains and amino acid residues are instrumental in this rDNA damage signaling pathway to induce the formation of PNAs.

**Reviewer #3 (Public Review):**
Summary:Hornofova et al. examined interactions between the nucleolus and promyelocytic leukemia nuclear bodies (PML-NBs) termed PML-nucleolar associations (PNAs). PNAs are found in a minor subset of cells, exist within distinct morphological subcategories, and are induced by cellular stressors including genotoxic damage. A systematic pharmacological investigation identified that compounds that inhibit RNA Polymerase 1 (RNAPI) and/or topoisomerase 1 or 2A caused the greatest proportion of cells with PNA. A specific RAD51 inhibitor (R02) impacted the number of cells exhibiting PNAs and PNA morphology. Genetic double-strand break (DSB) induction within the rDNA locus also induced PNA structures that were more prevalent when non-homologous end joining (NHEJ) was inhibited.Strengths:PNA are morphologically distinct and readily visualized. The imaging data are high quality, and rDNA is amenable to studying nuclear dynamics. Specific induction of rDNA damage is a strong addition to the non-specific pharmacological damage characterized early in the manuscript. These data nicely demonstrate that rDNA double-strand breaks undermine PNA formation. Figure 1 is a comprehensive examination and presents a compelling argument that RNAPI and/or TOP1, TOP2A inhibition promote PNA structures.Weaknesses:(1) The data are limited to fixed fluorescent microscopy of structures present in a minority of cells. Data are occasionally qualitative and/or based upon interpretation of dynamic events extrapolated from fixed imaging. This study would benefit from live imaging that captures PNA dynamics.

We fully agree with the reviewer that live-cell imaging is critical to adequately capture PNA formation and evolution dynamics. While the data presented in this manuscript are based on quantifications of fixed cell images, all these analyses are based on a detailed live-cell imaging examination of the dynamic behavior of PNAs that we reported in our orginal study on PNAs formation as a biological phenomenon Imrichova et al. (doi: 10.18632/aging.102248. Epub 2019 Sep 7).

In the revised version of our present manuscript, we better highlight the live-cell imaging study, in the Introduction section and further point out that the previous dynamic study was based on imaging of human cells ectopically expressing PML-EGFP and B23-RFP. Last but not least, to help the readers of this manuscript to understand the dynamics of PNA evolution, we have now also added an improved schematic figure that better illustrates the temporal dynamics of PNA stage transitions (Figure 1A).

(2) Cell cycle and cell division are not considered. Double-strand break repair is cell cycle dependent, and most experiments occur over days of treatment and recovery. It is unclear if the cultures are proliferating, or which cell cycle phase the cells are in at the time of analysis. It is also unclear if PNAs are repeatedly dissociating and reforming each cell division.

We agree that this is an important point. We previously published (Imrichova et al., doi: 10.18632/aging.102248) that exposure of RPE-1hTERT cells to doxorubicin caused cell cycle arrest and cellular senescence. In the revised manuscript, we added the analysis of how the I-PpoI-induced rDNA DSB affects the cell’s fate (Supplementary Figure 4J-N). Importantly, we found that most of the cells after I-PpoI-induced rDNA DSB also developed cellular senescence, and only 1–3% of cells eventually recovered from such rDNA stress to the extent that they were able to form colonies in a colony-forming assay. Thus, at the time of analysis, most of the cells were non-proliferating.

Additionally, in the revised manuscript, we included an analysis of the dependence of PNA formation on specific cell cycle phases (see Figures 6E–I and Supplementary Figure 6C–E). Generally, we found that PNAs can be present in G1/S/G2. Nevertheless, the probability of occurrence in a particular cell cycle phase is affected by the type of treatment. For example, after I-PpoI-induced rDNA damage, the PNAs are primarily present in G1. In contrast, after the sole knockdown of RAD51 or TOP2A, the PNAs are present in S/G2 with higher probability.

(3) The relationship of PNA morphologies (bowl, funnel, balloon, and PML-NDS) also remains unclear. It is possible that PNAs mature/progress through the distinct morphologies, and that morphological presentation is a readout of repair or damage in the rDNA locus. However, this is not formally addressed.

The reviewer is indeed correct in his/her interpretation of the PNA morphologies as a readout of the dynamic fate of the rDNA lesion. As mentioned in our response to the previous point no. 2 raised by this reviewer (see above), we described the dynamic structural PNA transitions in our previous article (Imrichova et al., doi: 10.18632/aging.102248).

PNA progresses through distinct structures. Our results indicate that individual PNA subtypes are tied to specific processes. The PNA bowl-type is linked to the recognition of rDNA damage on the nucleolar periphery. The PNA funnel-type clusters several damaged rDNA loci from the nucleolus into PML-NDS, which is the ultimate structure that sequesters unrepaired rDNA away from the reactivated nucleolus.

The formation of bowls, funnels, and balloons is linked to the inhibition of RNA polymerase I during the formation of nucleolar caps. In contrast, the later stage of PML-NDS is linked to RNA polymerase I reactivation.

We should mention that after the I-PpoI treatment, the ‘bowls’ and ‘funnels’ (observed originally in response to topoisomerase inhibitory drugs) are missing, and only PML-NDSs are formed. The apparent absence of the preceding stages of PNAs may reflect the lower extent of rDNA damage induced by I-PpoI treatment, without causing the pan-nucleolar RNA polymerase I inhibition that was observed for other treatments, such as doxorubicin.

(4) An I-Ppol targeted sequence within the rDNA locus suggests 3D structural rearrangement following damage. An orthogonal approach measuring rDNA 3D architecture would benefit comprehension.

This is a very inspiring idea. Given the demanding nature of the required 3D analyses and the fact that this aspect is somewhat outside the scope of the present study, we plan to follow this issue up in our future work, along with our efforts to localize the individual NORs using immune-FISH after introducing the rDNA damage by I-PpoI.

(5) Following I-Ppol induction, it is possible that cells arrest in a G1 state. This may explain why targeting NHEJ has a greater impact on the number of 53BP1 foci and should be investigated.

We fully agree with the Reviewer. Indeed, our results showed that after a 24-hour period of I-PpoI induction, most cells (about 90%) are in the G1 phase of the cell cycle, consistent with the activation of the ATM/ATR checkpoint signaling and p53 activation that we observed. Therefore, this cell cycle effect can indeed explain why targeting NHEJ has a greater impact and causes the higher numbers of 53BP1 foci (and also yH2AX foci).

(6) Conclusions: PNAs are a phenomenon of biological significance and understanding that significance is of value. More work is required to advance knowledge in this area. The authors may wish to examine the literature on APBs (Alt-associated PML-NBs), which are similar structures where telomeres associate with PML-NBs in a specific subset of cancers. It is possible that APBs and PNAs share similar biology, and prior efforts on APBs may help guide future PNA studies.

We are very grateful for this stimulating suggestion. In the Discussion of the revised manuscript, we now address the possible analogy between the APBs under ALT on the one hand, and the PNA formation on rDNA damage studied here, on the other. The following is the quote of the relevant paragraph of the revised Discussion:

“There are several similarities between PNAs and APBs. The interaction partner of PML located on both the telomeres and rDNA must be sumoylated, as the PML-SIM domain is essential for the formation of both APBs and PNAs (37,93). The PML IV isoform most efficiently forms APBs and also PNAs (16,37). PML clusters damaged telomeres into APBs, and we observe that several NORs converge in one PNA structure; thus, the PML-dependent clustering of damaged NORs is plausible. On the other hand, there is one critical difference between the otherwise broadly analogous APBs and PNAs. The process of ALT operates in transformed cancer cells that do not express the telomerase, thus enabling telomere maintenance, cell proliferation, and immortalization (94,95). The PNAs, on the other hand, were primarily detected in non-transformed cells, and their formation is linked to cell cycle arrest and establishment of senescence (31,36). It remains to be determined whether the formation of PNAs is positively involved in rDNA repair, resulting in a return of at least some PNA-forming cells to the cell cycle, or if they play a role in blocking the repair of DNA double-stranded breaks on rDNA, broadly analogous to the shelterin complex on telomeres during replicative senescence (96). We propose that the pro-senescent role of PNAs may contribute to the maintenance of rDNA stability, thereby limiting the potential of hazardous genomic instability and, hence, the risk of cellular transformation. Analogous to checkpoint responses and oncogene-induced senescence (97,98) the PNA-associated senescence might provide one aspect of the multifaceted cell-autonomous anti-cancer barrier, in this case guarding the integrity of the most vulnerable repetitive rDNA loci, possibly at the expense of accumulated cellular senescence-associated decline of functional tissues during aging.”

Our responses to recommendations from the Editors:(1) Since this paper does not provide a mechanistic insight into how the different PNA forms after DNA damage and PolI inhibition such as doxorubicin (DOXO) treatment and how HR modulates the PNA formation, it is very important to provide some experimental data for those. For example, as the #3 reviewer suggested, the time-lapse analysis of PML and a rDNA marker after DOXO treatment and recovery would be beneficial. with morphological analysis.

We fully agree that live-cell imaging is essential for a better understanding of the evolution and function of PNAs'. The requested time-lapse analysis on the dynamics of the PNA morphological stages after DOXO treatment and recovery is available to the Reviewers and readers in our previously published article that reported the PNA phenomenon and the basic live cell imaging data after doxorubicin treatment using the ectopically expressed PML-GFP and B23-RFP (Imrichova et al.; doi: 10.18632/aging.102248.). In our present revised manuscript, we now refer to this work in the Introduction and further stress that those data were based on live-cell imaging, to better highlight this point along the line recommended by the Reviewers. We have now also added an improved scheme that better explains the temporal dynamics of PNA transitions (Figure 1A).

(2) In the same line as point #1, it is very important to show what kind of signaling pathway is necessary for PNA formation upon DSB formation with PolI inhibition. For example, as the #2 reviewer advised, the role of ATM or ATR could be tested by adding their inhibitor during the PNA formation.

Again, we fully agree that clarification of the signaling pathway required for PNA formation is crucial, and we are grateful for this stimulating recommendation. While the mentioned Reviewer no. 2 (in his/her Public comments) asked only about the role of ATM, the Editors rightly requested that we should use distinct inhibitors to test the respective roles of not only ATM but also ATR. As recommended, we have tested the importance of ATM and ATR kinase activities by inhibiting them during PNA formation. These newly generated data clearly showed that the activity of either kinase is essential for the efficient formation of PNA, thereby providing a significant new mechanistic insight in the revised dataset. In the manuscript, these new results are now shown in Figures. 5A and B. We also addressed this issue in the Public Review (Reviewer #2 point 3).

(3) Given the association of PML body with telomeres in ALT cells (ALT-associated PML Body, APB) has been established well in the field, the authors need to mention this in the Introduction and also compare how PNA is similar to different from APB clearly in the Discussion.

We have followed this conceptually important recommendation exactly as suggested: (i) We now mention the ALT-associated PML Body (APB) in the Introduction section (end of the second paragraph) and (ii) In much more detail, we now compare the conceptual analogy in terms of similarities and differences between PNA and APB in the revised Discussion. We also address this issue in the document Response to Public Review (Reviewer #3 point 6). Indeed, we agree that this comparison is very fitting in the context of our dataset and informative for the broad audience.

**Recommendations for the authors:**

**Reviewer #1 (Recommendations For The Authors):**

Major points.(1) Any treatments shown in Figure 1B and 1C did not induce PNA in most of the cells with around 20% for a maximum value. What time point(s) the authors checked should be stated in the main text or the legend clearly. The authors need to mention the kinetics of different PNA classes and/or doseresponse effects at least for doxorubicin and BMH-21. Or a cell-cycle stage effect should be analyzed and/or discussed given that HR is mainly operating in S and G2 phases.

Thank you for pointing this out. We have now clarified the dose effects and also both analyzed and discussed the PNA formation vis a vis cell cycle stages, as recommended by this insightful reviewer.

First, we have now added an experimental scheme to the Figures for better clarity regarding the time points examined, as suggested.

Second, our results show that drug doses indeed affect the number and subtype of PNAs that form after such treatments. We show PNAs (types and number) after 0.5 – 5 – 50 µM camptothecin, topotecan, and etoposide (Supplementary Figure 1G and H) and after 0.375 – 0.56 – 0.75 µM doxorubicin (Figure 2A-D and Supplementary Figure 2E-G).

The very first detailed analysis of PNA evolution was presented in Imrichova et al. (doi: 10.18632/aging.102248.), where we described, using live-cell imaging, the relationship between the individual doxorubicin-induced PNA types, their transitions, and dynamics. We found that the highest number of nuclei with PNAs was present between 24 and 48 h after treatment initiation. Thus, we selected this time point for PNAs detection after treatments presented in Figure 1B.

We have now also added the distribution of nuclei based on the presence of specific PNA types into Supplementary Figure 1F.

We included the analysis of the dependence of PNA formation on specific cell cycle phases (see Figures 6E–I). A very detailed explanation of the observed cell cycle effects is presented in the document Responses to Public Review, re. Reviewer nr. 2, point 2, so please kindly read our response there.

(2) Although the induction of PNA by DSBs at rDNA repeats is clearly shown in the paper and modulated by DSB repair pathways, the biological significance of this sub-nuclear structure has not been addressed at all. Is the PNA required for efficient DSB repair per se or pathway choice? Moreover, the PNA kinetic is peculiar. Once formed, the PNA did not show any turnover even after the DNA-damaging agents were washed away (Figure 4H). This structure is succeeded into the next generation after cell division. Such dynamics of PNL should be carefully addressed.

The reviewer is correct in that the fate of the PNA and the potential biological significance of this phenomenon required a better explanation. The majority (≈97%) of cells after I-PpoI induction undergo cellular senescence, and therefore, we suppose that the PNA structures are not passed into the next cell cycle, as the bulk of the cells do not proliferate/cycle after such treatments. In this regard, it should be noted that PNAs (PML-NDS) are associated with replicative senescence of human mesenchymal stem cells (our old publication: Janderova-Rossmeislova 2007; doi: 10.1016/j.jsb.2007.02.008). To answer the comment of this reviewer, we have actually never observed that the cells with PNA present would be able to enter mitosis. Based on these findings, we suggest that damage to the repetitive rDNA loci, such as in our experiments in the form of DSBs, could commonly result in unsuccessful repair attempts leading to cellular senescence due to rDNA damage signaling, consistent with our new experiments highlighting the key role of the signaling mediated by the major DNA damage response kinases ATM and ATR, including the role of PNAs formation. For more details, please see also our response to Point 2 raised by the editors, on page 1 of this document, as well as our Public review response to Referee nr. 2, his/her points 2 and 3.

From a broader perspective, relevant to the biological function of PNAs in this unorthodox cellular stress response, we showed that doxorubicin-induced PML-NDSs separate/sequester persistent rDNA DSBs from the regions of active pre-rRNA transcription. Again, the purpose of this process is not entirely clear at present. However, such separation of unrepaired rDNA from the rest of the genome could have a protective function, thereby limiting the risk of aberrant homologous recombination among hundreds of the repetitive, recombination-prone rDNA copies spread across five chromosomes. It should be stressed that PNAs are rarely seen in cancer cells, and their absence might be linked to the rDNA instability commonly seen in transformed cells.

As published in our previous study (Imrichova et al.; doi: 10.18632/aging.102248.), we followed the fate of individual PML-NDS (the last stage of PNA) after the recovery from doxorubicin treatment using live-cell imaging. We observed that the destiny of this structure could be diverse. Some of them sustained in the nucleus for many hours, but a portion of them disappeared. Their extinction may be a manifestation of successful rDNA repair. However, what remains unresolved is why these cells do not reenter the cell cycle and instead develop a senescent phenotype, possibly reflecting some paracrine effects of a cocktail of diverse cytokines and chemokines secreted by the neighboring cells, a phenomenon well established in the senescence field as SASP (senescence-associated secretory phenotype).

Notably, during the recovery phase from I-PpoI insult, some of the PML-NDS, in fact, increase in size over time (please refer to the graph in Author response image 1). This enlargement suggests ongoing processes within these structures. Additionally, the sequential accumulation of DHX9 (a multifunctional DNA/RNA helicase) in PNAs during recovery from the I-PpoI insult (as shown in Figure 4G and Supplementary Figure 4H in the revised manuscript) supports the hypothesis that PNAs are associated with as-yet poorly understood process(es).

**Author response image 1. sa2fig1:** A scatter plot shows the changes in PNA diameters during the recovery phase from a 24-hour-long expression of IPpoI.

Last but not least, again relevant for the potential biological role of PNAs, we now also discuss the partial analogy of these structures with the PML-association with telomeres in cells that maintain their telomeres by the ALT recombinational process, as suggested by Referee no. 3 in the public review process. As this consideration addresses also the biological significance of the diverse PML associations and particularly our thoughts about the PNA, we copy/paste this paragraph from the Discussion section of our revised manuscript here, for the convenience of the Reviewer:

“There are several similarities between PNAs and APBs. The interaction partner of PML located on both the telomeres and rDNA must be sumoylated, as the PML-SIM domain is essential for the formation of both APBs and PNAs (37,93). The PML IV isoform most efficiently forms APBs and also PNAs (16,37). PML clusters damaged telomeres into APBs, and we observe that several NORs converge in one PNA structure; thus, the PML-dependent clustering of damaged NORs is plausible. On the other hand, there is one critical difference between the otherwise broadly analogous APBs and PNAs. The process of ALT operates in transformed cancer cells that do not express the telomerase, thus enabling telomere maintenance, cell proliferation, and immortalization (94,95). The PNAs, on the other hand, were primarily detected in non-transformed cells, and their formation is linked to cell cycle arrest and establishment of senescence (31,36). It remains to be determined whether the formation of PNAs is positively involved in rDNA repair, resulting in a return of at least some PNA-forming cells to the cell cycle, or if they play a role in blocking the repair of DNA double-stranded breaks on rDNA, broadly analogous to the shelterin complex on telomeres during replicative senescence (96). We propose that the pro-senescent role of PNAs may contribute to the maintenance of rDNA stability, thereby limiting the potential of hazardous genomic instability and, hence, the risk of cellular transformation. Analogous to checkpoint responses and oncogene-induced senescence (97,98) the PNA-associated senescence might provide one aspect of the multifaceted cell-autonomous anti-cancer barrier, in this case guarding the integrity of the most vulnerable repetitive rDNA loci, possibly at the expense of accumulated cellular senescence-associated decline of functional tissues during aging.”

(3) The association of PNA with DSB repair is shown by the co-localization with 53BP1 (Figures 3-5) and the kinetics of DSB repair were assessed by 53BP1 kinetics (Figure 5B). The authors need to check the co-localization of other DSB repair factors in homologous recombination (RPA and RAD51) and nonhomologous end joining (KU) and the kinetics of these DSB repair foci.

We are grateful for this very relevant suggestion. In response to this recommendation, we have examined additional markers, linked to homologous recombination. In Figures 6A—D and Supplementary Figures 6A and B, we now show also the localization of RAD51 and RPA32 (pS33), along the lines recommended by this Reviewer.

(4) In Figure 5B, 53BP1 foci in the "nucleolus" should be shown with that in the nucleus.

In the revised manuscript, we show histograms with a count of 53BP1 foci per nucleus.

(5) The authors often used the words, "difficult-to-repair" and "easy-to-repair" DNA lesions. However, without the nature of these DNA lesions, it is early to distinguish the lesions. So, the authors should avoid them in the title, abstract, results, and figure legends. In Discussion, it is free to use them with a logical explanation.

Thank you for the recommendation. We have now changed the term “difficult-to-repair” to “persistent rDNA damage”, as this term better describes at face value the scenario encountered in these experiments. In the new version of the manuscript, we have now emphasized that PNAs are formed as a late response to rDNA damage. We added the observation that PNAs co-localized with rDNA lesions accumulated in the nucleolar cap (periphery of nucleolus), which are probably in-compatible with NHEJ-mediated repair that otherwise occurs within the nucleolus. These persistent lesions contained phospho-RPA, a marker of resected DNA. However, RAD51 was not detected in such late lesions, indicating that the canonical RAD51-dependent HDR pathway is also restricted. Finally, we included a section defining such persistent DNA damage in the revised Discussion.

Minor points:(1) Page 5, second paragraph, line 6: "expression of PML".(2) Page 5, line 6 from the bottom and Figure 1B: Actinomycin D is not a "specific" RNA polymerase I inhibitor.(3) Page 6, first paragraph, last line: "DNA DSB" should be "DSB".(4) Page 6, second paragraph, lines 6-7: What is the evidence of RNA polymerase I is active (need to explain to the readers)?(5) Figure 1D and main text: Please mention DOXO is the abbreviation of doxorubicin.

We are grateful for these points, which have now all been corrected in the revised version of the manuscript.

(6) Page 6, third paragraph, line 4 and Figure 1D: What is "esi" not "si"TOP1.

In the revised manuscript, we explained what ‘esiRNA’ means; in fact, it is the pool of biologically prepared siRNAs targeting the mRNA of the protein being knocked down.

(7) Figures 2A and 2B: The effect of B02 alone on PNA should be shown as a control.

As recommended, the effect of B02 alone is now presented in Supplementary Figures 2A and B.

(8) Page 7, first paragraph, last three lines: It is hard to catch how the authors suggested the inhibition of RAD51 suppressed RNAPI activity. If so, please check the incorporation of 5FU.

Thank you for pointing out this confusing formulation. We have now removed from the revised manuscript the part of that original sentence: “which are predominantly associated with RNAPI inhibition”.

We observed that PML ‘balloons’ wrapped the nucleolus with the concomitantly observed complete inhibition of RNAPI in the nucleolus (Imrichova et al.; doi: 10.18632/aging.102248.). Nevertheless, we removed the original phrase from the revised version of the manuscript, as we agree with the reviewer that the causative relationship is so far lacking.

(9) Page 7, second paragraph: It is critical to clarify what time B02 was added after DOXO removal or during DOXO treatment, or both.

We agree: In response we have now added the experimental scheme showing all these temporal details.

(10) Figure 2H: The experiment lacks control with siTDP2 without etoposide treatment.

We did not include this control, unfortunately.

(11) Page 8, third paragraph, line 3 from the bottom; "besides of rDNA probe, we also utilized probes" is better.

We changed this sentence in the revised manuscript, as recommended.

(12) Figure 3B: In these multi-color images, it is hard to see blue and gray in merged ones. It is better to show images with a single color.

We agree that grayscale is better to follow. However, this type of presentation would significantly increase the number of images, a circumstance we wished to avoid in this already rather image-heavy dataset. Instead, when it was possible, we elevated the intensity of fluorescence in colored images. The list of images with this adjustment is present in the public review.

We also inserted the example of the image in greyscale here as Author response image 2.

**Author response image 2. sa2fig2:** The representative images nucleoli show the localization of 53BP1 (red; a marker of DNA DSB), PML (green, a marker of PML-NB or PNAs), rDNA (blue), and DJ (white; a marker of the acrocentric chromosome) after doxorubicin treatment (2 days) or in the recovery phase (1 and 4 days). The merge of all channels is shown together with the presentation of individual images in greyscale. Scale, 5 µm.

(13) Figure 4E: Please add values at D0.

We did not analyze the 53BP1 foci before adding Shield1 and doxycycline to induce the expression of I-PpoI (D0). However, as a control, we analyzed the 53BP1 foci in the cells treated for 24 h with the corresponding amount of DMSO as a mock treatment scenario (black line; NT).

**Reviewer #2 (Recommendations For The Authors):**
(1) The data provided in this manuscript did not explicitly compare the easy-to-repair vs difficult-torepair DNA lesions in rDNA, or at least lack quantitative measures with statistical analysis. Therefore, the title may need to be revised accordingly.

We agree, and the title has now been revised to better capture the persistent nature of the rDNA damage that evokes the PNA formation. Please see the response to Reviewer #1, Major points 5, presented above in this document.

**Reviewer #3 (Recommendations For The Authors):**
(1) Live imaging is paramount to understanding the dynamic nature of PNAs.

We agree that live-cell imaging is important. We have addressed this issue in detail in Response to Public review comments, of this Reviewer, as well as in the first point of this document in response to the Editors. In short, although the data presented in this manuscript are based on quantifications of fixed cell images, all these analyses benefit from our previous detailed live-cell imaging data that we reported – describing a careful examination of the dynamic behavior of PNAs in the study by Imrichova et al. (doi: 10.18632/aging.102248). To better illustrate the dynamic behavior of PNAs for the convenience of this reviewer, we include some data from our original article on this topic (referred to above): please see Author response image 3.

**Author response image 3. sa2fig3:** This Figure shows data published in Imrichova et al. (doi: 10.18632/aging.102248). PML IV-EGFP was ectopically expressed in RPE-1hTERT cells. The localization of PML was followed using live cell imaging. (A) The bowl (in this work named cap) originates from the accumulation of diffuse PML. (B) The transition between bowl (named cap), funnel (named fork), and balloon (named circle). (C, D) PML IV-EGFP (green) and B23-RFP (red) were ectopically expressed in RPE-1hTERT cells. The localization of both proteins was followed by live cell imaging. (C) The formation of PML-NDS from the funnel is shown; (D) The entire PNA cycle is shown. PML-bowl formed on the border of the nucleolus, then transformed into the PML-funnel, and finally into PML-NDS.

(2) The authors should consider cell cycle and cell proliferation in their analyses.

We are grateful for this recommendation, which echoes your own comment nr. 2 in the Public reviews document. Shortly, as we explained in the response to Public review, proliferation of PNA-containing cells is severely limited, as the vast majority of such cells enter a long-term arrest and cellular senescence. Furthermore, inspired by this comment, we have newly performed a series of experiments to address the frequencies of PNA formation vis a vis cell cycle phase position of the individual cells with rDNA damage. In the revised manuscript, we now include the data from these analyses: see Figures 6E–I and Supplementary Figures 6C–E. Our response in the Public Review provides a detailed description of these results.

(3) Merged fluorescent micrographs in red and green are potentially not discernible to individuals with colour-vision deficiencies. Consider re-colouring into schemes that are more accessible.

We agree that some readers may have different preferences about fluorescence micrographs. Here, we used the classical combination of green and red, commonly employed in the field.

(4) Single-colour fluorescent micrographs are easier to visualize in grey-scale. Whenever a single colour is shown, it will help reader comprehension if the images are shown in this manner.

As recommended, we have changed Figures 4C, F, and G from a single-color presentation to a greyscale.

(5) There are many long paragraphs that are difficult to digest. I suggest where possible breaking this text into smaller portions (e.g. Page 10, pages 13-14, page 16-17).

Thank you for pointing this out. We have now broken the text into smaller portions (in several places), as recommended.

(6) The B02 and NU7441 data would be bolstered by genetic confirmation (depleting RAD51, BRCA2 or PALB2 for HR, DNA-PK or LIG4 for NHEJ).

As recommended, we downregulated Rad51 and LIG4 by RNA interference. New data are presented in Figures 5F–I, 6E, and F, Supplementary Figures 5D, E, F–H, and Supplementary Figures 6C–E. The Public Review provides a detailed description of these results and the ensuing conclusions.

(7) Microscopy results are often qualitative (Fig S1I, S2L, S3A) and need to be bolstered with quantitative data.

We appreciate this recommendation and have implemented quantifications in several important microscopy results, as follow:

S1I: The quantification of the number of cells with types of PNAs after esiTOP1 is present in Supplementary Figure 1L

S2L: The quantification (% of nuclei with PNAs) is in Figure 2H

S3A: In this immuno-FISH figure, we captured nuclei with and w/o PNAs. Using the SQUASSH analysis, we identified size-based co-localization between rDNA–PML and DJ–PML presented in Supplementary Figure 3C.

(8) Stats or error bars are missing (Fig 1D, 2H, S1C-E, S1F, S2A S2D-G, S3E, S4E).

We apologize for those omissions and we have amended this aspect of the study in the revised manuscript as much as possible:

Figure 1D: For AMD and doxorubicin and CX-5461 and doxorubicin treatments, three and two biological replicates are shown separately in the same graph, respectively. For AMD and the knockdown of TOP1, the mean from three biological replicates is shown. All these results indicate the elevation number of PNAs when RNAPI is inhibited.

Figure 2H: The error bars are present. As for siTDP2 in all replicates, the number of cells was the same (4%). Therefore, the error bar is not visible.

Supplementary Figure 1C-E: Unfortunately, only one replicate (for all treatments) was analyzed by western blotting.

Supplementary Figure 1F (in revised manuscript SF1G): The error bars are present. By this graph, we mainly wanted to present the variation in PNAs types.

Supplementary Figure 2A (in revised manuscript SF2C): We include the whiskers plot (box shows 10-90 percentile and line median) and T-test.

Supplementary Figure 2D-G (in revised manuscript SF2F-I): The error bars are present in all graphs. The changes in SF2F and G are not significant.

Supplementary Figure 3E: This scheme shows the overlaps between rDNA and PML and rDNA and 53BP1. The collum graph based on these data is shown in Figure 3F.

Supplementary Figure 4E: The plot profiles representing the mean fluorescence of PML and B23 are shown for different time points.

(9) PNA characteristics remind this reviewer of the well-described ALT-associated PML nuclear bodies (APBs) found in immortalized cells lacking telomerase (i.e. Alternative lengthening of telomeres). I recommend the authors look to published data on APBs to help guide how to approach their research within a framework of the cell cycle.

We fully agree with this insightful comment, and have addressed this point in the Discussion section of the revised manuscript, quoted the relevant studies also in the Introduction, and indeed explained the parallels and also differences of PNA versus APB (see also our response to point 3 highlighted also by the Editors, early in this rebuttal document). We have also addressed this issue in the Public Review (Reviewer #3 point 6). We agree with the reviewer that this comparison will be of wide interest to readers, given the potential insights into the biological roles of APBs and PNAs.

For convenience, we copy/paste the relevant new paragraph of the Discussion here:

“There are several similarities between PNAs and APBs. The interaction partner of PML located on both the telomeres and rDNA must be sumoylated, as the PML-SIM domain is essential for the formation of both APBs and PNAs (37,93). The PML IV isoform most efficiently forms APBs and also PNAs (16,37). PML clusters damaged telomeres into APBs, and we observe that several NORs converge in one PNA structure; thus, the PML-dependent clustering of damaged NORs is plausible. On the other hand, there is one critical difference between the otherwise broadly analogous APBs and PNAs. The process of ALT operates in transformed cancer cells that do not express the telomerase, thus enabling telomere maintenance, cell proliferation, and immortalization (94,95). The PNAs, on the other hand, were primarily detected in non-transformed cells, and their formation is linked to cell cycle arrest and establishment of senescence (31,36). It remains to be determined whether the formation of PNAs is positively involved in rDNA repair, resulting in a return of at least some PNA-forming cells to the cell cycle, or if they play a role in blocking the repair of DNA double-stranded breaks on rDNA, broadly analogous to the shelterin complex on telomeres during replicative senescence (96). We propose that the pro-senescent role of PNAs may contribute to the maintenance of rDNA stability, thereby limiting the potential of hazardous genomic instability and, hence, the risk of cellular transformation. Analogous to checkpoint responses and oncogene-induced senescence (97,98) the PNA-associated senescence might provide one aspect of the multifaceted cell-autonomous anti-cancer barrier, in this case guarding the integrity of the most vulnerable repetitive rDNA loci, possibly at the expense of accumulated cellular senescence-associated decline of functional tissues during aging.”

(10) Do PNAs mature/progress through the four distinct structures: bowl, to funnel, to balloon, and finally to PML-NDS. If true, this serves as a phenotypic read-out of damage induction (bowl) and repair (PML-NDs). It would suggest persistent unrepairable damage (0.56 or 0.75 uM doxorubicin) prevents repair leading to the formation of all the PNA structures except PML-NDs. While lower dose doxorubicin (0.375 uM) allows repair to occur, facilitating progression to the PML-ND state, which is then inhabited with B02.

Again, this is a very insightful comment. Indeed, as the Reviewer suggests and as we explained e.g., in our response to point 1 raised by this reviewer, PNA progresses through four distinct structures/maturation stages. Our results indicate that individual PNA subtypes are tied to specific processes. PNA bowl-type is linked to the recognition of rDNA damage on the nucleolar surface. The PNA of the funnel-type clusters several rDNA loci from the nucleolus into PML-NDS, which is the ultimate structure sequestering unrepaired rDNA away from the reactivated nucleolus.

There is a negative correlation between doxorubicin dose and occurrence of PML-NDS, and, indeed, blocking HDR with BO2 combined with a lower doxorubicin dose results in a higher occurrence of all PNAs, including PML-NDS, emerged in the recovery phase. These findings indicate that the greater/more severe extent of rDNA damage, which is associated with RNAPI activity inhibition, is linked to PNAs types associated with RNAPI inhibition originally published Imrichova et al. (doi: 10.18632/aging.102248.). In contrast, a milder degree of rDNA damage induces the formation of PMLNDS.